# Drep-2 is a novel synaptic protein important for learning and memory

Till F M Andlauer[1,2,3], Sabrina Scholz-Kornehl[1], Rui Tian[1,2], Marieluise Kirchner[4], Husam A Babikir[1], Harald Depner[1], Bernhard Loll[5], Christine Quentin[1], Varun K Gupta[1], Matthew G Holt[6], Shubham Dipt[7], Michael Cressy[8], Markus C Wahl[5], André Fiala[7], Matthias Selbach[4], Martin Schwärzel[1], Stephan J Sigrist[1,9]*

[1]Genetics, Institute of Biology, Freie Universität Berlin, Berlin, Germany; [2]Rudolf Virchow Center, DFG Research Center for Experimental Biomedicine, Julius-Maximilians-Universität Würzburg, Würzburg, Germany; [3]Max Planck Institute of Colloids and Interfaces, Potsdam, Germany; [4]Department of Cell Signalling and Mass Spectrometry, Max-Delbrück-Centrum für Molekulare Medizin, Berlin-Buch, Germany; [5]Institute of Chemistry and Biochemisty, Freie Universität Berlin, Berlin, Germany; [6]Department Laboratory of Glia Biology, Vlaams Instituut voor Biotechnologie (VIB) Center for the Biology of Disease, Katholieke Universiteit Leuven, Leuven, Belgium; [7]Department of Molecular Neurobiology of Behavior, Georg-August-Universität Göttingen, Göttingen, Germany; [8]Department of Neuroscience, Cold Spring Harbor Laboratory, Cold Spring Harbor, United States; [9]NeuroCure Cluster of Excellence, Charité Universitätsmedizin Berlin, Berlin, Germany

*For correspondence: stephan.sigrist@fu-berlin.de

**Abstract** CIDE-N domains mediate interactions between the DNase Dff40/CAD and its inhibitor Dff45/ICAD. In this study, we report that the CIDE-N protein Drep-2 is a novel synaptic protein important for learning and behavioral adaptation. Drep-2 was found at synapses throughout the *Drosophila* brain and was strongly enriched at mushroom body input synapses. It was required within Kenyon cells for normal olfactory short- and intermediate-term memory. Drep-2 colocalized with metabotropic glutamate receptors (mGluRs). Chronic pharmacological stimulation of mGluRs compensated for *drep-2* learning deficits, and *drep-2* and *mGluR* learning phenotypes behaved non-additively, suggesting that Drep 2 might be involved in effective mGluR signaling. In fact, *Drosophila* fragile X protein mutants, shown to benefit from attenuation of mGluR signaling, profited from the elimination of *drep-2*. Thus, Drep-2 is a novel regulatory synaptic factor, probably intersecting with metabotropic signaling and translational regulation.

## Introduction

Caspase family proteases regulate cellular pathways by cleavage of target proteins. They are best known for their roles during programmed cell death. One of their targets is the DNase Dff40/CAD, which degrades DNA during apoptosis (*Enari et al., 1998*). Dff40 is a member of the DNA fragmentation factor (Dff) family of proteins, characterized by CIDE-N domains that mediate protein–protein interactions (*Wu et al., 2008*). In *Drosophila*, four CIDE-N domain proteins were identified and named Dff related protein Drep-1 to Drep-4 (*Inohara and Nuñez, 1999*). Caspase-regulated Drep-4 is the ortholog of mammalian Dff40/CAD (*Yokoyama et al., 2000*). Drep-4 is inhibited by Drep-1, the ortholog of Dff45/ICAD, which is also cleaved by caspases (*Mukae et al., 2000*). The two other *Drosophila* family members, Drep-2 and -3, were suggested to be additional regulators of apoptosis, solely based on in vitro interactions (*Inohara and Nuñez, 1999*; *Park and Park, 2012*).

**eLife digest** Synapses are specialized structures that connect nerve cells to one another and allow information to be transmitted between the cells. Synapses are essential for learning and storing memories. Many proteins that regulate how signals are transmitted at synapses have already been studied. In this manner, much has been learned about their function in learning and memory.

Cells can commit suicide by a process called apoptosis, also known as programmed cell death. Apoptosis is not only triggered in damaged cells but is also necessary for an organism to develop correctly. In fruit flies, the protein Drep-2 is a member of a family of proteins that degrade the DNA of cells that undergo apoptosis.

Andlauer et al. found no evidence that Drep-2 plays a role in apoptosis, but have now found Drep-2 at the synapses of the brain of the fruit fly *Drosophila*. Drep-2 could be observed in close proximity to another type of protein called metabotropic glutamate receptors. Metabotropic glutamate receptors and their signaling pathways are important for regulating certain changes to the synapses that mediate learning processes. Indeed, Andlauer et al. found that flies that have lost the gene that produces Drep-2 were unable to remember smells when these were paired with a punishment. Stimulating the regulatory glutamate receptors with drugs helped to overcome learning deficits that result from the lack of Drep-2.

Alterations in the production of a protein called FMRP cause fragile X syndrome in humans, the most common form of hereditary mental disability originating from a single gene defect. Flies lacking the FMRP protein show learning deficits that are very similar to the ones seen in flies that cannot produce Drep-2. However, Andlauer et al. observed that flies lacking both Drep-2 and FMRP can learn normally.

Exactly how Drep-2 works in synapses to help with memory formation remains to be discovered, although there are indications that it boosts the effects of signaling from the glutamate receptors and counteracts FMRP. Further research will be needed to establish whether the mammalian proteins related to Drep-2 perform similar roles in the brains of mammals.

We show here that Drep-2, contrary to the expectations, is a novel synaptic protein. We have generated *drep-2* mutants, which display learning and memory deficits. Further analyses suggest that Drep-2 regulates these processes by intersecting with metabotropic glutamate receptor signaling.

## Results

*Drosophila* encodes two Dff40/CAD-related proteins, Drep-4 and Drep-2 (*Inohara and Nuñez, 1999*; *Yokoyama et al., 2000*) (*Figure 1—figure supplement 1*). Both proteins are controlled by caspase-mediated cleavage of their inhibitors Drep-1 and Drep-3, respectively (*Mukae et al., 2000*; *Park and Park, 2012*, *2013*). The roles of Drep-4 and -1 during apoptosis have been firmly established, yet the functions of Drep-2 and -3 have so far remained unknown (*Inohara and Nuñez, 1999*; *Park and Park, 2012*; *Tan et al., 2012*).

### Drep-2 is a novel synaptic protein without a discernable role in the regulation of apoptosis

The gene coding for Drep-2 is located on chromosome IIR and contains five exons (*Figure 1A*). While the last two exons are used in all known isoforms, the first three exons are included alternatively within the four isoforms drep-2-RA to -RD.

High-throughput RT-PCR data indicate a strong enrichment of both *drep-2-RA* and *drep-3* transcripts in the central nervous system (CNS), while *drep-4* and *-1* are expressed ubiquitously (*Graveley et al., 2011*). We conducted in situ hybridization using *drep-2-RA* constructs in order to confirm that the expression of *drep-2* is nervous system specific (*Figure 1B*). Next, we produced polyclonal antibodies against a fusion protein containing the C-terminal half of the protein, which is part of all isoforms (Drep-2$^{C-Term}$). Western blots from wild-type fly head extracts probed with Drep-2$^{C-Term}$ showed a double band (*Figure 1C*) of the size expected for Drep-2 isoforms (*McQuilton et al., 2012*).

We generated *drep-2* mutants by FLP-mediated excision between FRT site-bearing transposons to explore the function of Drep-2. The transposons P(XP)d00223 and PBac(RB)e04659 were used

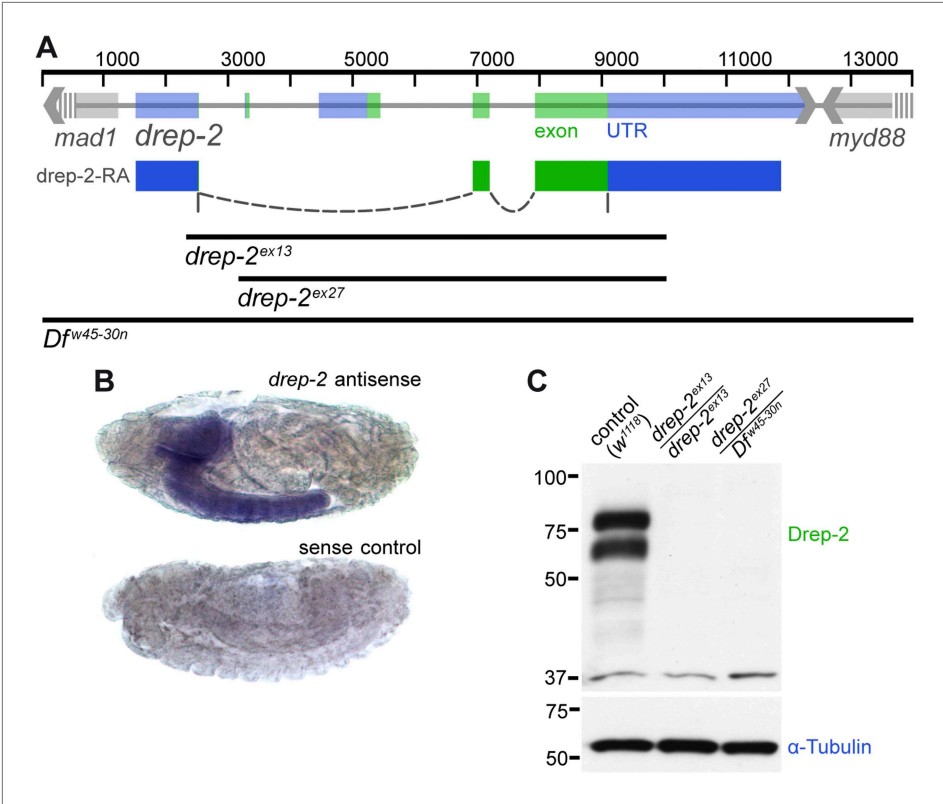

**Figure 1**. Expression and mutants of *drep-2*. (**A**) Genetic scheme of the *drep-2* locus on chromosome IIR. The neighboring genes *mad-1* and *myd88* extend beyond the sequence displayed. The cDNA labeled *drep-2-RA* was used for rescue experiments. Blue: untranslated regions; green: exons; black lines: deleted regions in the mutants. (**B**) In situ hybridization of *drep-2* reveals a neuronal expression pattern (stage 17). (**C**) Western blot of adult fly head extracts using the anti-Drep-2$^{C-Term}$ antibody. Drep-2 isoforms are predicted to run at 52 and 58 kDa. The signal is absent in both the *drep-2$^{ex13}$* and the *drep-2$^{ex27}$/Df$^{w45-30n}$* mutant.

The following figure supplements are available for figure 1:

**Figure supplement 1**. Drep protein alignment.

**Figure supplement 2**. Reduced lifespan of *drep-2$^{ex13}$* mutants.

for the deletion allele *drep-2$^{ex13}$* (**Figure 1A**). A second deletion allele, *drep-2$^{ex27}$*, was established using transposon lines PBac(RB)e02920 and PBac(RB)e04659. All *drep-2* exons are deleted in homozygous *drep-2$^{ex13}$* animals, while no other annotated transcription unit is affected. In the smaller intragenic deletion *drep-2$^{ex27}$*, all *drep-2* exons apart from the very small (12 bp) first exon are eliminated. Both Drep-2$^{C-Term}$ antibody bands were absent in head extracts of both mutants (**Figure 1C**), confirming the complete elimination of Drep-2 expression in these deletion alleles. Flies lacking *drep-2* were viable and fertile but shorter-lived than the isogenic controls (**Figure 1—figure supplement 2**).

Subsequently, we used the Drep-2$^{C-Term}$ antibody for wholemount immunostainings of *Drosophila* brains. The synaptic neuropil was strongly labeled throughout the brains of larvae (*not shown*) and adults (**Figure 2**). In both *drep-2* mutants, Drep-2$^{C-Term}$ staining in the CNS was completely abolished (*drep-2$^{ex13}$*: **Figure 2A,B**).

As a Dff family member, Drep-2 has been suggested to be involved in the apoptotic regulation of DNA degradation (***Inohara and Nuñez, 1999***; ***Park and Park, 2012***). However, neuronal cell bodies lacked Drep-2$^{C-Term}$ staining (**Figure 2**). We produced synaptosome-like preparations by fractionation of the adult *Drosophila* CNS to biochemically confirm the association of Drep-2 with synapses (***Owald et al., 2012***; ***Depner et al., 2014***). Drep-2 was strongly enriched in fractions containing

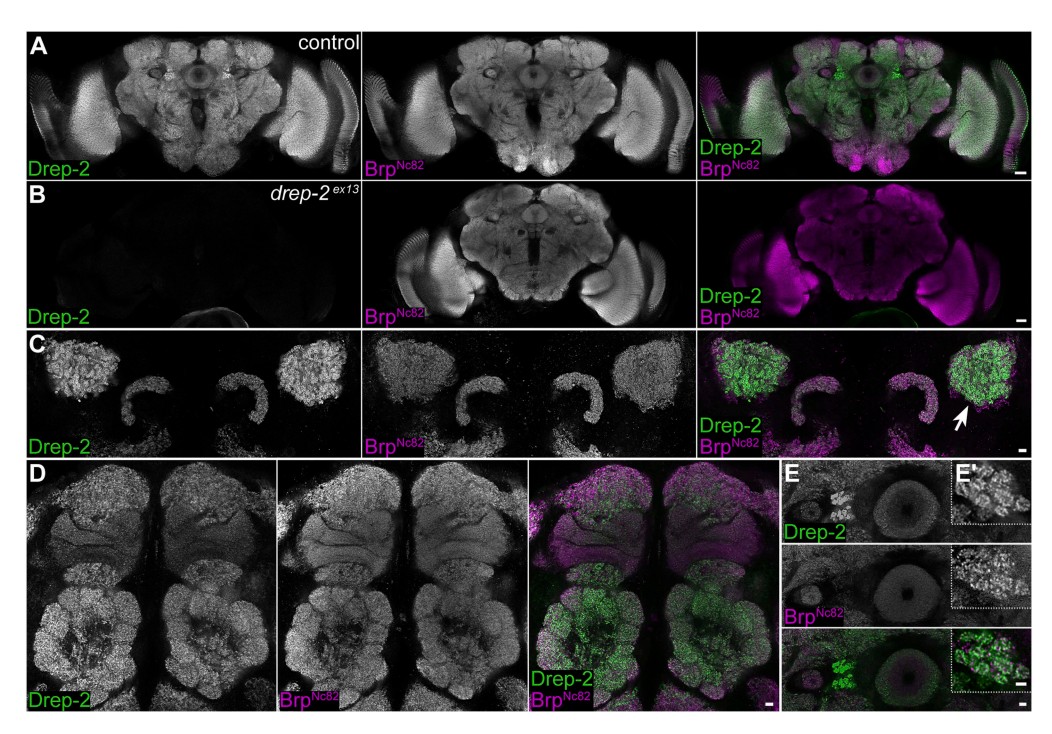

**Figure 2**. Synaptic Drep-2 staining in the CNS. (**A**–**B**) Confocal frontal sections of adult *Drosophila* brains. Anti-Drep-2$^{C-Term}$ and Brp$^{Nc82}$ immunostaining; the latter marks all synaptic active zones. Synaptic Drep-2$^{C-Term}$ signal is visible throughout the brain of wild-type flies (**A**). Complete loss of the anti-Drep-2$^{C-Term}$ staining can be observed in *drep-2$^{ex13}$* mutants (**B**). Scale bars: 20 μm. (**C**–**E**) Frontal sections of wild-type brains, anti-Drep-2$^{C-Term}$, and Brp$^{Nc82}$ staining. Scale bars: 5 μm. (**C**) Posterior–dorsal detail showing strong Drep-2 staining in MB calyces (arrow). (**D**) Anterior frontal section with antennal lobes and MB lobes. (**E**) Ellipsoid body in the central complex and bulbs (lateral triangles) (**E'**: magnification of strong Drep-2 staining in bulbs).

synaptic membranes (*Figure 3A*). By contrast, no enrichment of Drep-2 could be observed in the nuclear fraction.

Mutants lacking *drep-2* showed normal exterior morphology, including their facet eyes (*Figure 3B*). Fly facet eyes are highly ordered structures, already displaying abnormal patterns in cases of moderate misregulation of apoptosis (*Wolff and Ready, 1991*; *Song et al., 2000*). The regular facet eyes of *drep-2* mutants, therefore, argue against a major function of the protein in apoptosis. Wholemount brain stainings of wildtypes and *drep-2$^{ex13}$* mutants also showed no apparent morphological differences between either genotype (*Figure 2A,B*; *Figure 5—figure supplement 1*). If apoptosis was, nevertheless, misregulated in the CNS of *drep-2* mutants in vivo, an altered count of cell bodies should be expected in adult flies. Drep-2 staining was especially prominent at KC synapses in the mushroom body (MB) calyx of wildtypes (*Figure 2C*). We, therefore, quantified the numbers of cell bodies of a subset of MB-intrinsic neurons (KCs). No differences between *drep-2* mutants and controls could be observed (*Figure 3C*).

Drep-2 was reported to degrade DNA in vitro (*Park and Park, 2012*). This supposed nuclease activity was observed if purified Drep-2 was incubated in vitro with plasmid DNA at a molar ratio of protein:DNA 80:1 (*Park and Park, 2012*). However, we (*Figure 3D*) and another previous report (*Inohara and Nuñez, 1999*) found no evidence of a nuclease activity of Drep-2, even at high concentrations. Instead, Drep-2 appeared to precipitate DNA, as evident by plasmid DNA no longer entering the agarose gel when incubated with Drep-2. This precipitation might have generated the previous impression that DNA is degraded in the presence of Drep-2. Taken together, we could not find evidence for an in vivo role of Drep-2 in regulating apoptosis in the CNS. While a function of the protein related to apoptosis can still not be fully excluded, we decided rather to examine the synaptic functions of Drep-2.

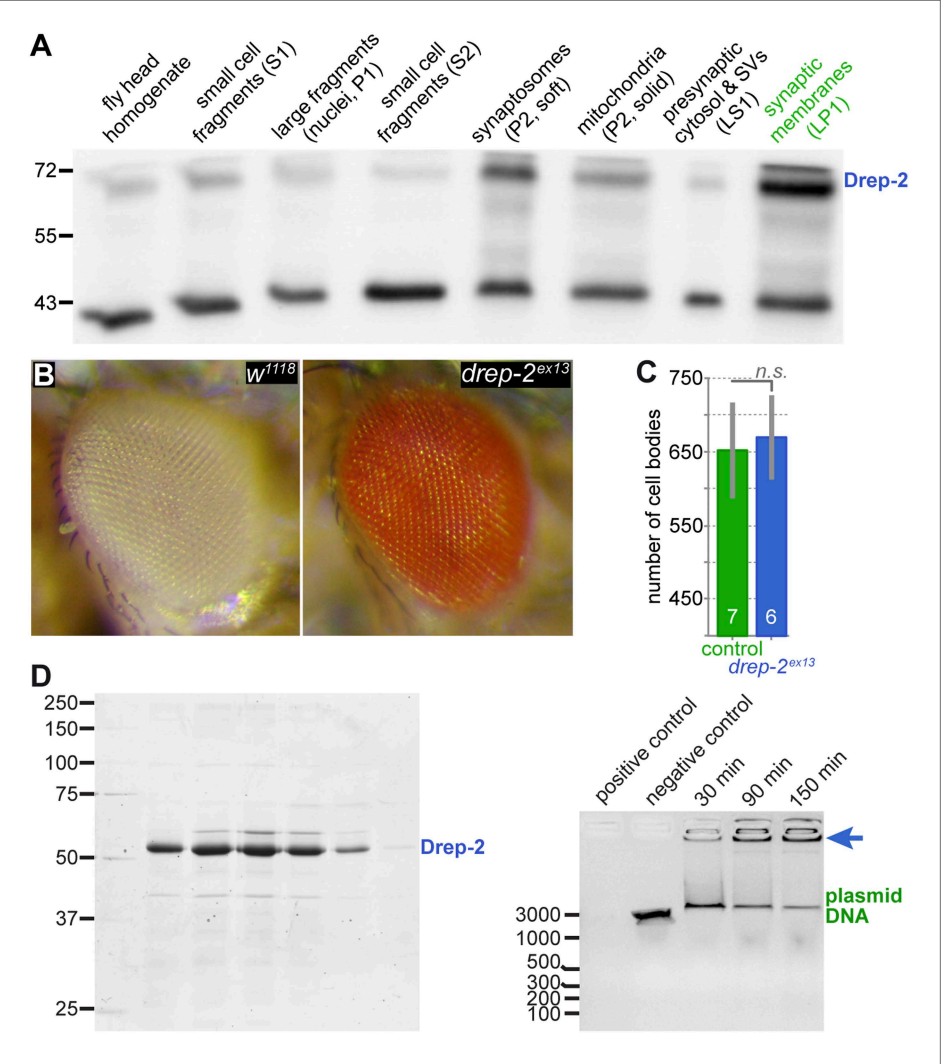

**Figure 3**. No evidence for a role of Drep-2 in regulation of apoptosis. (**A**) Synaptosome-like preparation of adult wild-type head extracts (***Depner et al., 2014***), probed with Drep-2[C-Term]. Drep-2 is concentrated in fractions containing synaptic membranes. S = supernatant, P = pellet, L = (after) lysis. Please see the protocol by ***Depner et al. (2014)*** for a more detailed explanation of the fractionation procedure. (**B**) Mutants (*drep-2[ex13]*) did not show a rough eye phenotype. The facet eyes of flies, highly ordered structures, are often affected in apoptosis mutants. By contrast, the eyes of *drep-2* mutants appeared normal. (**C**) The number of mb247-positive KCs does not differ between *drep-2[ex13]* mutants and controls. GFP was expressed using the MB KC driver mb247-Gal4. GFP-positive cell bodies were counted and compared between genotypes. No significant difference was found between mean cell body counts (Mann–Whitney U test, p = 0.886). Average cell body counts were in the expected range: control = 651, mutant = 669, published = 700 (***Schwaerzel et al., 2002***). (**D**) Purified Drep-2 does not degrade linearized plasmid DNA. Left: SDS-PAGE of the final elusion profile of purified Drep-2, loaded onto a HighLoad Superdex S200 16/60 column. Right: Nuclease activity assay of purified Drep-2 analyzed by 1% (wt/vol) agarose gel. Drep-2 was incubated in a time course experiment with linearized plasmid DNA. No nuclease activity could be detected. Instead, Drep-2 seemed to precipitate DNA, as evidenced by high-molecular DNA not entering into the agarose gel when incubated with Drep-2 (arrow).

## Drep-2 is strongly enriched at the postsynaptic densities of mushroom body input synapses

In order to narrow down a potential site of action of the protein, we examined Drep-2 expression in the adult CNS in more detail. While Drep-2[C-Term] stained synapses throughout the brain, including optic lobes, antennal lobes, and the central complex (***Figure 2***), the immunoreactivity was particularly pronounced in the MB calyx (***Figures 2C*** and ***Figure 4A***).

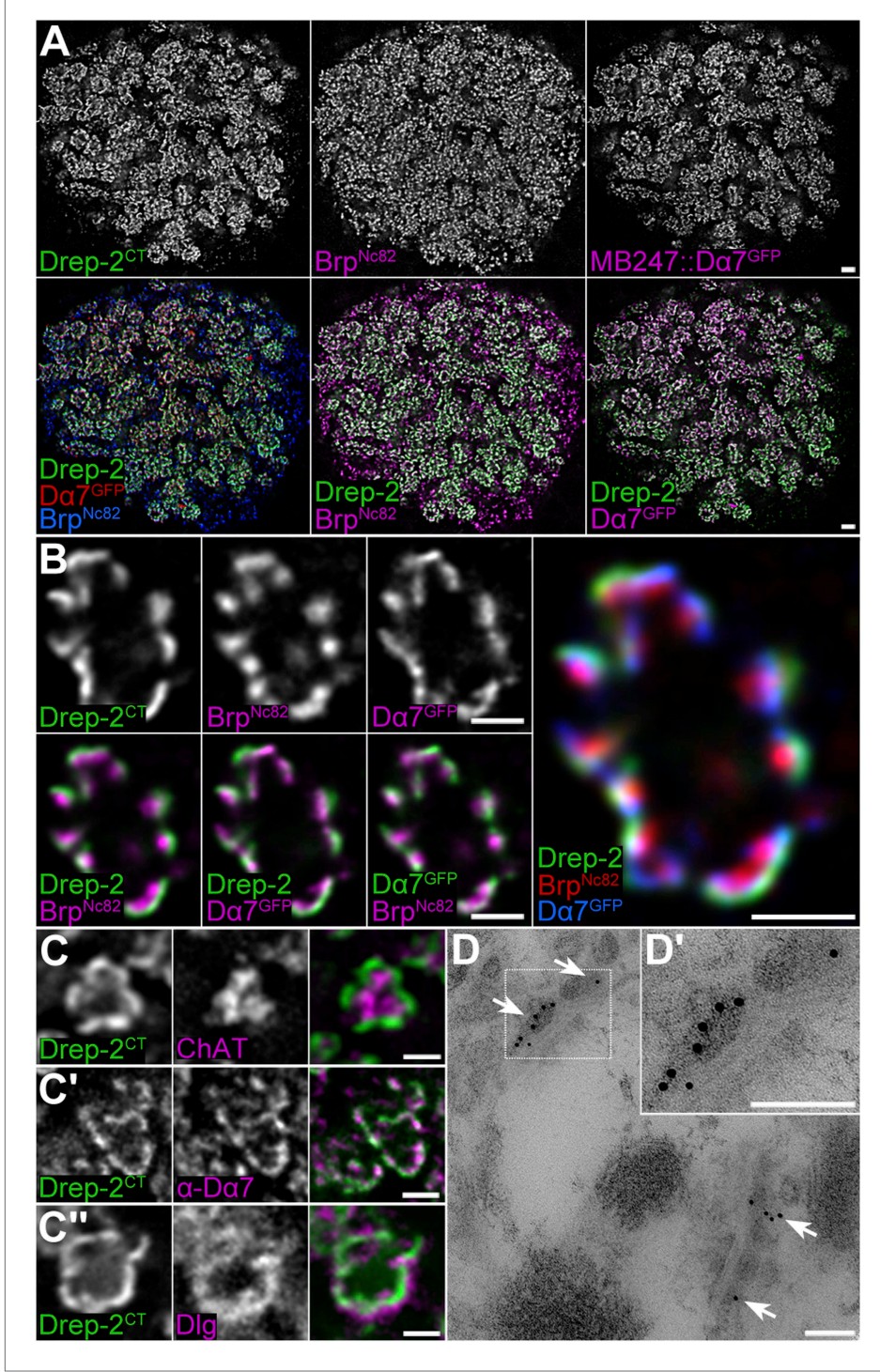

**Figure 4**. Drep-2 is enriched at KC postsynapses. (**A–B**) Drep-2[C-Term] and Brp[Nc82] staining in animals expressing the construct mb247::Dα7[GFP] that marks acetylcholine receptors in MB KCs. (**A**) Detailed image of the MB calyx. Scale bar: 2 µm. (**B**) Detail of a single microglomerulus in the calyx. Drep-2[C-Term] overlaps with postsynaptic mb247::Dα7[GFP] and not with presynaptic Brp. Scale bars: 1 µm. (**C**) Localization of Drep-2 relative to choline acetyltransferase (ChAT, presynaptic cytosol, C), the postsynaptic ACh receptor subunit Dα7 (antibody staining, **C'**), and the postsynaptic scaffolding protein Discs large (Dlg, **C''**). Drep-2 colocalizes with postsynaptic markers.
*Figure 4. Continued on next page*

*Figure 4. Continued*

Scale bars: 1 µm. (**D**) Post-embedding immunoelectron microscopy of Drep-2$^{C\text{-Term}}$ in the calyx. Arrows: Clusters of postsynaptic Drep-2$^{C\text{-Term}}$. Scale bars: 100 nm.

The following figure supplement is available for figure 4:

**Figure supplement 1**. Drep-2 localizes to postsynaptic membranes of KCs in the calyx.

---

The principle circuitry processing olfactory information in *Drosophila* is highly similar to mammals (*Davis, 2004*). Signals are transferred in the fly's antennal lobe to projection neurons (PNs), which target the mushroom body (MB) calyx and the lateral protocerebrum. PNs synapse in the calyx onto MB-intrinsic Kenyon cells (KCs) by forming large cholinergic presynaptic boutons. These presynaptic specializations are tightly encircled by acetylcholine (ACh) receptor-expressing postsynaptic densities (PSDs) of KC dendritic claws (*Yasuyama et al., 2002*; *Leiss et al., 2009a*).

In the MB calyx, the Drep-2$^{C\text{-Term}}$ signal overlapped with Dα7 ACh receptor subunits expressed in KC PSDs (*Figure 4A,B,C'*), which surrounded Bruchpilot (Brp)-positive (but Drep-2-negative) PN presynapses (*Figure 4A,B*, *Figure 4—figure supplement 1A*). Furthermore, Drep-2 colocalized with the postsynaptic protein Discs large, but clearly segregated from presynaptic choline acetyltransferase (*Figure 4C*). Consistent with endogenous Drep-2 being present at KC-derived PSDs, overexpression of UAS-Drep-2$^{mStrawberry}$ with either pan-neural (elav$^{c155}$-Gal4) or KC-specific drivers (c305a-Gal4) resulted in an mStrawberry signal equivalent to that of the Drep-2$^{C\text{-Term}}$ antibody (*Figure 4—figure supplement 1B,C*). Overexpression with a PN driver (gh146-Gal4), however, produced only a weak, diffuse expression pattern that bore no similarity to endogenous Drep-2 staining (*Figure 4—figure supplement 1D*). Moreover, re-expression of UAS-Drep-2 in KCs of *drep-2* mutants produced a distinctive label at KC PSDs (*Figure 4—figure supplement 1E*). Finally, we confirmed the presence of Drep-2 at postsynaptic membranes of PN-KC synapses by immunoelectron microscopy (*Figure 4D*). Thus, Drep-2 accumulates at postsynaptic specializations of KCs within the MB calyx.

## Drep-2 is required in KCs for normal olfactory short- and intermediate-term memory

Kenyon cells have an essential function in olfactory learning (*Davis, 2011*; *Dubnau and Chiang, 2013*). Here, conditioned stimuli (odors) get associated with unconditioned stimuli (e.g., electric foot shock). Because Drep-2 was strongly enriched at PSDs of KCs, we wondered whether the protein might contribute to olfactory learning. In aversive olfactory conditioning, flies are trained to learn the difference between a punished and an unpunished odor. We measured short-term memory (STM) directly following training and intermediate-term memory (ITM) after 3 hr. All flies used for behavioral experiments were outcrossed into our isogenic background ($w^{1118}$) for at least five generations. Performance of controls allowed for the detection of differences in either direction (*Figure 5B,C*).

Mutants lacking *drep-2* showed normal naïve sensory acuity, with innate olfactory responses and shock avoidance behavior not significantly different from controls (*Figure 5A*). However, the STM performance scores of both mutants (*drep2$^{ex13}$* and *drep2$^{ex27}$*) were significantly lower than the scores of isogenic $w^{1118}$ controls (*Figure 5B,C*). Re-expression of *drep-2* cDNA by using either the pan-neural driver elav$^{III}$-Gal4 or two different KC-specific drivers (30y-Gal4 and mb247-Gal4) was sufficient to rescue STM scores (*Figure 5C*). Drep-2 is, therefore, required in KCs for normal olfactory STM.

ITM is composed of two memory components, consolidated anesthesia-resistant memory (ARM) and labile anesthesia-sensitive memory (ASM) (*Quinn and Dudai, 1976*; *Tully et al., 1994*). ASM and ARM rely on different molecular and neuronal mechanisms (*Folkers et al., 1993*; *Tully et al., 1994*; *Scheunemann et al., 2012*). Amnestic cooling abolishes the labile ASM and is, thus, used to separate both components. The *drep-2* mutants exhibited regular ARM but were deficient in ASM (*Figure 5D*), which was evidenced by memory scores not differing between cooled and untreated *drep-2* mutants. This loss of ASM was rescued by re-expression of *drep-2* using elav$^{III}$-Gal4 or KC-specific mb247-Gal4. We conclude that Drep-2 is required in KCs for both STM and ASM, but not for ARM.

Concurrently, morphologies of *drep-2*-mutant MBs and PN-KC synapses appeared normal (*Figure 5—figure supplement 1*). Furthermore, the number of synapses in the MB calyx did not differ between either genotype (*Figure 5—figure supplement 1D*). Thus, *drep-2*-mutant phenotypes

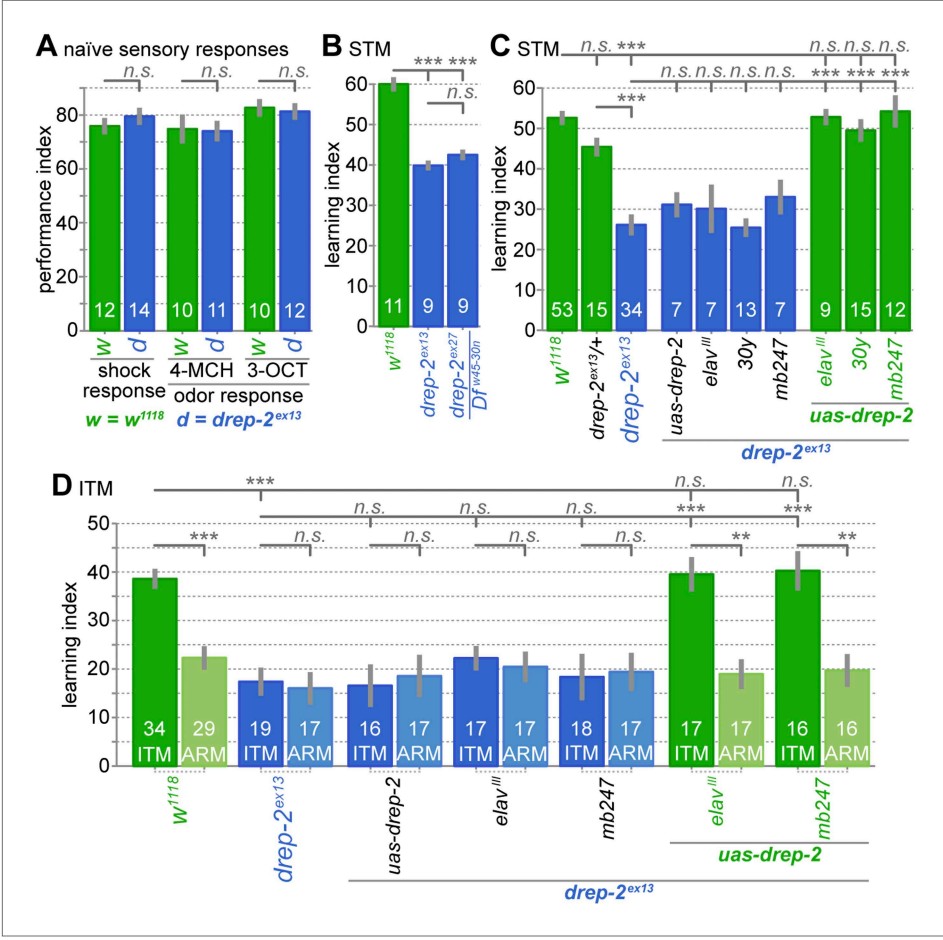

**Figure 5**. Drep-2 is required in KCs for olfactory short- and intermediate-term memory. (**A**) Flies mutant for *drep-2* sense electric shock and the odors 4-methyl-cyclohexanol (4-MCH) and 3-octanol (3-OCT) normally; there is no difference in mean performance indices between mutants and isogenic *w1118* control flies (Mann–Whitney U tests (MWU)). Sample sizes n are indicated with white numbers; grey bars show SEMs. (**B**) Both mutants *drep-2ex13* and *drep-2ex27/Dfw45-30n* are deficient in aversive olfactory conditioning, 3 min STM in a T-maze. The graph shows mean learning indices and SEMs. Mutants performed significantly worse than isogenic controls (MWU: p = 0.00001 for both comparisons, Bonferroni-corrected significance level α = 0.0167, 3 tests). (**C**) Re-expression of *drep-2* cDNA with elav III-Gal4 (pan-neural), 30y-Gal4 (MB KCs), or mb247-Gal4 (MB KCs) restores the deficit to normal levels. Heterozygous *drep-2ex13* mutants do not display a significant deficit. MWU for individual comparisons showed a significant difference between these groups (α = 0.0042, 12 tests): *w1118* and *drep-2ex13* (p < 0.00001), *drep-2ex13/drep-2ex13* and *drep-2ex13/+* (p < 0.00001), *drep-2ex13* and *drep-2ex13;uas-drep-2/elavIII-gal4* (p < 0.00001), *drep-2ex13* and *drep-2ex13;uas-drep-2/30y-gal4* (p < 0.00001), *drep-2ex13* and *drep-2ex13;uas-drep-2/mb247-gal4* (p < 0.00001). None of the differences indicated as not significant had a p < 0.12, except for *w1118* and *drep-2ex13/+* (p = 0.03851; not significant in the case of α = 0.0042). (**D**) Intermediate-term memory (ITM = ASM + ARM) performance. Mutants (*drep-2ex13*) are defective in ASM, but not in ARM. The defect can be restored with elav III-Gal4 or mb247-Gal4 (30y-Gal4 was not used here). Statistical tests were run separately for ITM and ARM. For ITM, MWU for individual comparisons showed a significant difference between these groups (α = 0.00625, 8 tests): *w1118* and *drep-2ex13* (p < 0.0001), *drep-2ex13* and *drep-2ex13;uas-drep-2/elavIII-gal4* (p < 0.0001), *drep-2ex13* and *drep-2ex13; uas-drep-2/mb247-gal4* (p < 0.0001). For assessing differences in ARM, ITM and ARM performances of each genotype were compared with MWU. The following genotypes showed a significant difference between ITM and ARM (α = 0.0071, 7 tests): *w1118* (p < 0.0001), *drep-2ex13;uas-drep-2/elavIII-gal4* (p = 0.0002), *drep-2ex13; uas-drep-2/mb247-gal4* (p = 0.0006). None of the differences indicated as not significant had a p < 0.11.

The following figure supplement is available for figure 5:

**Figure supplement 1**. PN-KC synapses appear morphologically normal in *drep-2* mutants.

should not be caused by gross structural developmental aberrations. Instead, given the synaptic localization of the protein, it appeared likely that Drep-2 might intersect with signaling pathways involved in synaptic plasticity.

## Functional overlap between Drep-2 and mGluR in olfactory conditioning

PN-KC synapses, as is the case for many excitatory synapses in the *Drosophila* CNS, use ACh as the main fast neurotransmitter (*Gu and O'Dowd, 2006*; *Yasuyama et al., 2002*). Drep-2 colocalized here with nicotinic ACh receptor subunits (*Figure 4*). In addition, several types of metabotropic receptor are also expressed in the calyx, including GABA$_B$, dopamine, octopamine, and metabotropic glutamate receptors (*Enell et al., 2007*; *Devaud et al., 2008*; *Busch et al., 2009*; *Mao and Davis, 2009*; *Kanellopoulos et al., 2012*). DmGluRA is the single functional metabotropic glutamate receptor (mGluR) in *Drosophila* (*Parmentier et al., 1996*), orthologous to mammalian group II/III mGluRs. DmGluRA and the mGluR-associated protein Homer show a characteristic expression throughout the *Drosophila* brain, with strong expression in the MB calyx (*Ramaekers et al., 2001*; *Diagana et al., 2002*; *Hamasaka et al., 2007*; *Urizar et al., 2007*; *Devaud et al., 2008*; *Kahsai et al., 2012*; *Kanellopoulos et al., 2012*).

We examined different receptor types and related proteins for colocalization with Drep-2 (*not shown*). Drep-2 colocalized with both mGluR and Homer throughout the brain (*not shown*), with very prominent co-labeling in the MB calyx (*Figure 6A*). This made mGluR signaling a prime candidate for a pathway interacting with Drep-2 function.

Recently, mGluRs have been found to be important for olfactory conditioning in flies. In that work, a decrease of mGluR levels provoked by RNA interference improved performance scores of the olfactory learning mutant *dfmr1* (*Kanellopoulos et al., 2012*). The same effect was observed upon administration of the mGluR antagonist 2-methyl-6-(phenylethynyl)pyridine (MPEP).

We assayed the STM performance of the *mGluR* mutant *dmGluRA$^{112b}$* to independently confirm a function of mGluRs in olfactory learning (*Bogdanik et al., 2004*). Indeed, a significant reduction in learning scores was observed in these mutants (*Figure 6B*). In addition to the mGluR antagonist MPEP (*McBride et al., 2005*; *Bolduc et al., 2008*; *Tauber et al., 2011*; *Kanellopoulos et al., 2012*), the agonist 1S,3R-1-amino-1,3-cyclopentanedicarboxylate (ACPD) has also been shown to be effective in flies (*Parmentier et al., 1996*; *Hamasaka et al., 2007*). We, therefore, tested the effects of both MPEP and ACPD on olfactory learning scores of *drep-2* mutants (*Figure 6C*). To this end, we raised flies on food containing either of the two components throughout development and adulthood. In agreement with our result for *dmGluRA* mutants (*Figure 6B*), the antagonist (MPEP) significantly decreased learning scores of wild-type flies when fed throughout development (*Figure 6C*). By contrast, MPEP did not further reduce the learning ability of *drep-2* mutants. However, feeding the mGluR agonist ACPD during development effectively rescued the *drep-2$^{ex13}$* phenotype (*Figure 6C* (*+ACPD dev+ad*)). At the same time, feeding the agonist to controls did not alter their learning scores. When ACPD was fed only to adult animals after eclosion, it had no discernible effect on the performance of *drep-2* mutants (*Figure 6C* (*+ACPD adult*)). In summary, artificial activation of mGluR receptors starting during development can compensate for the olfactory learning deficits of *drep-2* mutants.

In order to further investigate a potential relationship between Drep-2 and mGluRs, we produced *drep-2; dmGluRA* double mutants. These double mutants showed learning indices that were very similar to the scores of both single mutants (*Figure 6D*). As the learning deficits of both mutants did not add up to a stronger impairment in double mutants, it is likely that both proteins converge, at least partially, into a common regulatory pathway.

## Loss of *drep-2* antagonizes *dfmr1* phenotypes

Activation of mGluR signaling is thought to stimulate local synaptic translation through a signaling cascade involving Homer. By contrast, the RNA-binding fragile X mental retardation protein FMRP was shown to repress translation, thereby counteracting mGluR-mediated synaptic translation (*Bhakar et al., 2012*). Loss of FMRP function causes fragile X syndrome (FXS), the most frequent monogenic intellectual disorder. Pharmacological treatment with allosteric inhibitors of mGluRA was demonstrated to attenuate phenotypic deficits in rodent models of FXS and in FXS patients (*Krueger and Bear, 2011*; *Bhakar et al., 2012*; *Gross et al., 2012*).

In *Drosophila*, FMRP function and mGluR signaling also behave antagonistically. Importantly, learning phenotypes of *dfmr1* mutants lacking FMRP can be rescued by pharmacological *inhibition*

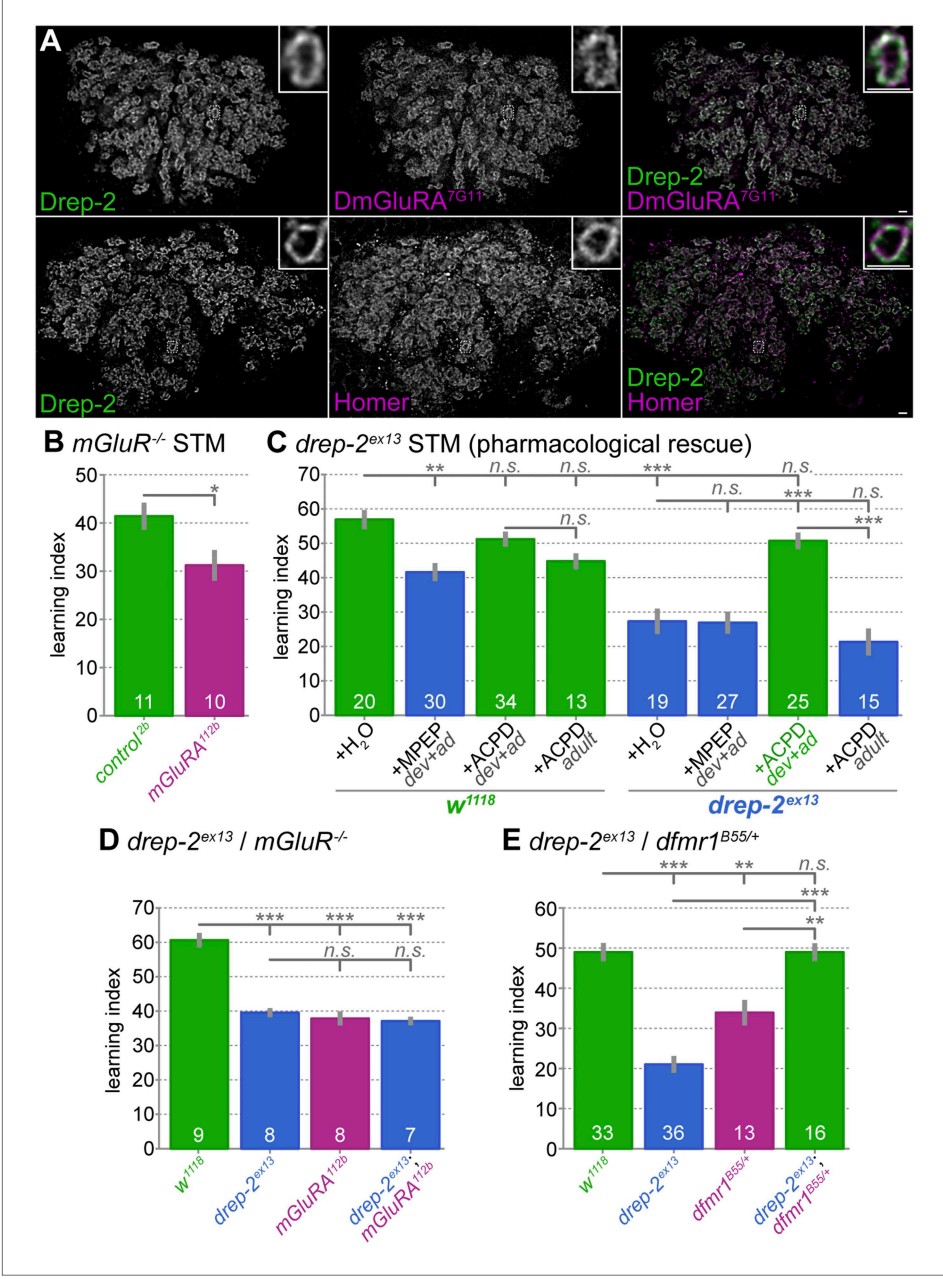

**Figure 6**. Functional overlap between Drep-2 and mGluR in olfactory conditioning. (**A**) Wildtype adult MB calyces stained with Drep-2[C-Term] and DmGluRA[7G11] (first row) or with Drep-2[C-Term] and anti-Homer (second row). Drep-2 colocalizes tightly with both proteins. The insets show single microglomeruli. Scale bars: 2 μm. (**B**) Flies carrying the mutation *dmGluRA[112b]* are deficient in aversive olfactory conditioning STM when compared to isogenic *dmGluRA[2b]* controls that do express DmGluRA; MWU: p = 0.043, α = 0.05. The graph shows mean learning indices and SEMs; sample sizes n are indicated with white numbers. (**C**) The *drep-2[ex13]* phenotype in olfactory STM can be rescued by raising animals on food containing the DmGluRA agonist 1S,3R-ACPD (ACPD). Food was supplemented throughout development and adulthood with either the DmGluRA receptor antagonist MPEP (9.7 μM) or the agonist ACPD (72.2 μM) diluted in $H_2O$ (label: *dev+ad*). Control animals received only $H_2O$. One group of animals was transferred to food supplemented by ACPD only after eclosion and not during development; the corresponding experiments are indicated by the label *+ACPD adult*. MPEP lowered the *w[1118]* performance significantly (MWU p = 0.0003). MPEP did not alter *drep-2[ex13]* indices (p = 0.8772) and ACPD did not change *w[1118]* performance (p = 0.1145). ACPD rescued the *drep-2* mutant phenotype to control levels if fed during both development and adulthood (comparison of *drep-2[ex13]* +ACPD dev+ad to untreated *drep-2[ex13]*: p < 0.00001; comparison to untreated

*Figure 6. Continued on next page*

*Figure 6. Continued*

$w^{1118}$: p = 0.0945). ACPD did not rescue the mutant phenotype if fed only during adulthood (*+ACPD adult*, no significant difference to untreated *drep-2$^{ex13}$* (p = 0.2281), significant difference to mutants treated with ACPD during both development and adulthood (p < 0.00001)). The difference between untreated *w$^{1118}$* and *drep-2$^{ex13}$* flies was also significant (p < 0.00001). Significance level α = 0.005 (10 tests). (**D**) Phenotypes of *drep-2$^{ex13}$; dmGluRA$^{112b}$* double mutants were non-additive. Both *drep-2$^{ex13}$* and *dmGluRA$^{112b}$* single mutants showed significantly lower olfactory STM than isogenic controls (MWU, p = 0.00008 for both comparisons). Double mutants showed similar learning indices (comparison to *w$^{1118}$*: p = 0.00018). The two single mutants and the double mutant did not significantly differ from each other (p > 0.178). α = 0.0083 (6 tests). (**E**) Loss of *drep-2* antagonizes *dfmr1* phenotypes in olfactory conditioning STM. Both homozygous *drep-2$^{ex13}$* mutants and heterozygous *dfmr1$^{B55}$/+* mutants are deficient in olfactory learning STM, but double mutants carrying both alleles do learn. The graph shows mean learning indices and SEMs. MWU for individual comparisons (α = 0.01, 5 tests): *w$^{1118}$* and *drep-2$^{ex13}$* p < 0.00001, *w$^{1118}$* and *dfmr1$^{B55}$/+* p = 0.00069, *w$^{1118}$* and *drep-2$^{ex13}$; dfmr1$^{B55}$/+* p = 0.83751, *drep-2$^{ex13}$* and *drep-2$^{ex13}$; dfmr1$^{B55}$/+* p < 0.00001, *dfmr1$^{B55}$/+* and *drep-2$^{ex13}$; dfmr1$^{B55}$/+* p = 0.00071.

of mGluRs (**McBride et al., 2005**; **Bolduc et al., 2008**; **Tauber et al., 2011**; **Kanellopoulos et al., 2012**). By contrast, we describe here that *drep-2* mutants profited from pharmacological *stimulation* of mGluRs (**Figure 6C**). We wondered, therefore, whether *drep-2* and *dfmr1* mutants would behave antagonistically.

To this end, we generated *drep-2; dfmr1* double mutants. Single and double *dfmr1* mutants lacking both copies of FMRP (*dfmr1$^{B55}$/dfmr1$^{Δ50M}$*) only hatched in small numbers, insufficient for olfactory conditioning experiments. However, **Kanellopoulos et al. (2012)** have shown that heterozygous *dfmr1* mutants are also deficient in olfactory learning. We, therefore, compared homozygous *drep-2* single mutants to heterozygous *dfmr1$^{B55}$/+* single and *drep-2$^{ex13}$/drep-2$^{ex13}$; dfmr1$^{B55}$/+* double mutants regarding aversive olfactory conditioning performance. Both single mutants showed decreased olfactory learning (**Figure 6E**), confirming the published olfactory learning phenotype of *dfmr1/+* heterozygotes. Notably, however, performance of *drep-2; dfmr1/+* double mutants was indistinguishable from the controls (**Figure 6E**), despite the deficit of both single mutants. This suggests that the absence of Drep-2 functionally compensates for the loss of FMRP. Our experiments thus provide first evidence that Drep-2 and FMRP display a functional antagonism.

## Drep-2 and FMRP found in common complexes

Finally, we began exploring the molecular basis of the behavioral connections between Drep-2 and both mGluR signaling and FMRP. We first examined whether Drep-2 might regulate the protein levels or localization of either mGluR or Homer. However, mGluR and Homer levels appeared unaltered in *drep-2* mutants (*not shown*). Therefore, it appeared more likely for Drep-2 to intersect with signaling processes downstream of the mere metabotropic glutamate receptor complex. We used quantitative affinity purification in combination with mass spectrometry to learn about Drep-2 in vivo interaction partners (**Vermeulen et al., 2008**; **Paul et al., 2011**). Unfortunately, the Drep-2$^{C-Term}$ antibody did not precipitate the endogenous Drep-2 protein sufficiently to allow for such an analysis. Thus, a Drep-2$^{GFP}$ fusion protein showing a localization pattern identical to endogenous Drep-2 (**Figure 7—figure supplement 1**) was expressed pan-neurally using elav$^{c155}$-Gal4. We isolated protein complexes of this pan-neurally overexpressed Drep-2$^{GFP}$ from fly heads using anti-GFP beads (**Figure 7**). Parallel pulldowns, using either plain beads or Drep-2$^{GFP}$-negative lysates, were performed as controls for non-specific binding (**Figure 7A**). All three pulldowns were conducted in triplicate and processed and analyzed by high-resolution shotgun proteomics. Proteins were quantified by label-free quantification and specific interaction partners were extracted using t-test statistics (**Hubner et al., 2010**).

On top of the bait proteins Drep-2 and GFP (the latter because the fusion protein Drep-2$^{GFP}$ was expressed), 35 proteins were robustly enriched over both controls at a false-positive discovery rate (FDR) of 1% (**Figure 7B**, **Supplementary file 1**). In order to visualize which of these 35 core proteins are part of a larger grid of putative interactors, an extended protein network was generated (**Figure 7D**). Proteins that DroID (**Murali et al., 2011**) or Flybase (**McQuilton et al., 2012**) lists as (putative) interactors of any of the 35 core proteins (FDR 1%) were added if they fitted the following three conditions: (i) enriched in the pulldowns (elav$^{c155}$; uas-drep-2$^{GFP}$ flies, GFP beads vs plain beads) at an FDR of 10%, (ii) not enriched in the control experiment (GFP beads, wild-type flies vs elav$^{c155}$; uas-drep-2$^{GFP}$ flies) at

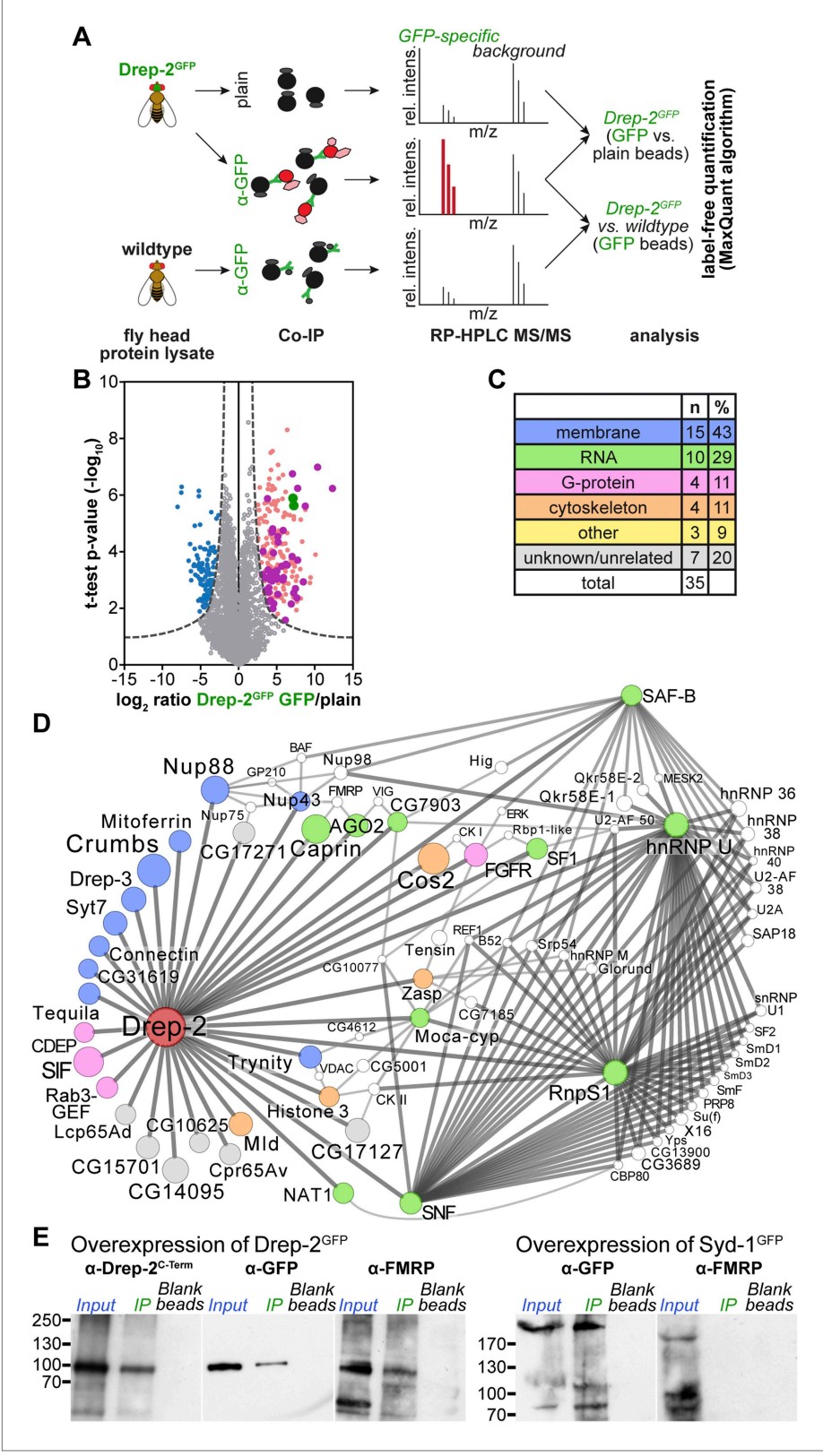

**Figure 7**. Quantitative mass spectrometry: Drep-2 and FMRP were found in a common protein complex.
(**A**) Strategy for the identification of Drep-2 interactors by quantitative mass spectrometry. UAS-Drep-2$^{GFP}$ was overexpressed using the pan-neural driver line elav$^{c155}$-Gal4. (**B**) Volcano plot showing proteins from Drep-2$^{GFP}$ flies

*Figure 7. Continued*

binding to anti-GFP and/or plain control beads. A hyperbolic curve (set at an FDR of 1%) separates GFP-enriched proteins (light pink) from background (grey). Proteins enriched in the control are shown in blue. Proteins that were significantly enriched, both in Drep-2$^{GFP}$ flies and in independent control experiments with wild-type flies, are colored magenta (n = 35). Drep-2 and GFP are shown as green dots. (**C**) Classification of the 35 core network proteins; multiple counts were allowed. (**D**) Network of the 35 proteins that were significantly and reproducibly enriched in GFP pulldown experiments (at an FDR of 1%, magenta-colored dots in **B**). Additional putative interactors of the core network (FDR set at 10%) are shown in white (*Supplementary file 2*). The circle (node) and font size correspond to the rank within the results (indicated in *Supplementary files 1 and 2*). The line (edge) width and shade correspond to the number of interactions each of the significantly enriched proteins has with others. The line/edge length is arbitrary. (**E**) Anti-FMRP probing confirmed the specific presence of FMRP in Drep-2$^{GFP}$ complexes. Head extracts of flies expressing Drep-2$^{GFP}$ or the presynaptic protein Syd-1$^{GFP}$ were processed in parallel. FMRP was only enriched in preparations of Drep-2$^{GFP}$ extracts. Immunoprecipitations were performed using either GFP-Trap-A beads (lanes labeled *IP*) or blocked agarose beads as binding control (labeled *Blank beads*).

The following figure supplement is available for figure 7:

**Figure supplement 1**. Drep-2$^{GFP}$ colocalizes with endogenous Drep-2.

an FDR of 10% (to eliminate false-positives), and (iii) a (predicted) interaction with at least two of the 35 core proteins (*Supplementary file 2*).

Among the proteins found to be significantly enriched were Drep-2, GFP, and Drep-3, a cognate binding partner of Drep-2 (*Inohara and Nuñez, 1999*; *Park and Park, 2012*). Thus, we were successful in precipitating proteins interacting with Drep-2 and not merely peptides binding unspecifically. 14 of the 35 putative core interacting proteins are associated with membranes (*Figure 7C*, *Supplementary file 1*), consistent with our observation that Drep-2 localizes to the postsynaptic plasma membrane. However, neither DmGluRA nor Homer could be identified in the preparation.

Both mGluR and FMRP regulate local synaptic translation (*Bhakar et al., 2012*). Interestingly, 10 of the 35 proteins have been implicated in the control of mRNA translation and stability. Network analysis of putative interacting proteins (*Figure 7D*) underlined a strong connection of Drep-2 with RNA-associated proteins as well. Among them, several proteins regulating translation were found: for example, the eIF4G-related, cap-independent translation initiation factor NAT1/p97/DAP5 (*Levy-Strumpf et al., 1997*; *Hundsdoerfer et al., 2005*) and Caprin, a dendritic translational repressor (*Shiina et al., 2005*). Notably, Argonaute-2, involved in RNA interference (*Ketting, 2011*), was also among the most highly enriched proteins (*Supplementary files 1 and 2*). It is interesting in this context that both Caprin and Argonaute-2 bind to FMRP (*Ishizuka et al., 2002*; *Papoulas et al., 2010*). Finally, FMRP was identified within Drep-2 complexes at an FDR cutoff of 5% (*Supplementary file 2*). In fact, we could directly confirm the presence of FMRP in Drep-2$^{GFP}$ immunoprecipitates by immunoblotting (*Figure 7E*). FMRP, by contrast, could not be detected in an identically treated control experiment conducted in parallel, in which we precipitated the presynaptic protein Syd-1$^{GFP}$ (*Owald et al., 2010*) (*Figure 7E*). Thus, FMRP appears to be present specifically within Drep-2$^{GFP}$ complexes.

Since Drep-2 could be detected in protein complexes containing FMRP, translational control processes constitute a possible place of action of Drep-2. However, further mechanistic analysis will have to work out the details of the regulatory functions executed by the novel synaptic protein Drep-2.

## Discussion

We here identify the CIDE-N family protein Drep-2 as a novel synaptic protein expressed in the *Drosophila* CNS (*Figure 2*) and important for learning and memory (*Figure 5*). Loss of Drep-2 did not cause transmission deficits at neuromuscular junctions and photoreceptor synapses (*not shown*) and we did not observe any structural deficits at synapses (*Figure 5—figure supplement 1*). Thus, Drep-2 is most likely less important for maintaining either base-line transmission or fundamental synaptic architecture. Instead, it might be involved in the regulation of synaptic signaling and plasticity.

Expression of Drep-2 is particularly strong at postsynaptic densities (PSDs) of synapses between projection neurons (PNs) and Kenyon cells (KCs) in the mushroom body (MB) calyx (*Figure 4*). At these synapses, acetylcholine is released from PNs upon transmission of odor signals (*Busto et al., 2010*; *Gu and O'Dowd, 2006*). While elimination of Drep-2 in the whole animal severely impaired olfactory

short-term and anesthesia-sensitive intermediate-term memory (*Figure 5*), re-expression of Drep-2 restricted to KCs was sufficient to fully rescue these learning deficits. As sensation of electric shock, the unconditioned stimulus in aversive olfactory learning, is mediated via dopamine in the MB lobes (*Aso et al., 2012*), Drep-2 most likely plays its role at PN::KC synapses during reception of the odor, the conditioned stimulus.

Several types of metabotropic receptors are expressed in the calyx: GABA$_B$, dopamine, octopamine, and metabotropic glutamate receptors (*Enell et al., 2007*; *Devaud et al., 2008*; *Busch et al., 2009*; *Mao and Davis, 2009*; *Kanellopoulos et al., 2012*). Which exact role this metabotropic signaling plays in synaptic plasticity processes at PN::KC synapses is essentially unknown. Motivated by the close match between Drep-2 and DmGluRA localization on the level of individual PSDs of the PN-KC synapse (*Figure 6A*), we started to address a potential functional connection between Drep-2 and DmGluRA-dependent signaling and behavior. We observed that the learning deficits of *dmGluRA* and *drep-2* single mutants did not add up to a stronger phenotype in double mutants (*Figure 6D*). Moreover, the *drep-2* memory deficit was effectively rescued by pharmacological stimulation of DmGluRA (*Figure 6C*). Drep-2 might, therefore, affect plasticity processes downstream of DmGluRA. In this pathway, the protein could be required for extracting relevant aspects of the olfactory information (odor discrimination vs generalization) and thereby influence subsequent learning. Interestingly, preliminary experiments indicate that the learning deficits of *drep-2* mutants might involve decreased Ca$^{2+}$ responses in KCs (*not shown*).

How might Drep-2 interfere with metabotropic signaling? We were unable to biochemically detect DmGluRA in complexes containing Drep-2, making a direct interaction between both proteins unlikely. Neither could we find any indication of an influence of Drep-2 on mGluR protein levels or localization (*not shown*). However, we did confirm the presence of the fragile X mental retardation protein FMRP in complexes containing Drep-2 (*Figure 7E*). In addition to FMRP, several other additional RNA-associated proteins and translational regulators were found by quantitative affinity purification experiments of Drep-2$^{GFP}$ complexes, followed by mass spectrometry-based protein identification and quantification (*Figure 7C,D*, *Supplementary files 1 and 2*). It is, therefore, a probable scenario that Drep-2 indirectly regulates local synaptic protein synthesis.

Several studies have demonstrated that FMRP antagonizes mGluR-mediated synaptic translation (*Bhakar et al., 2012*). Notably, we found evidence of a functional antagonism between Drep-2 and FMRP-mediated plasticity: both *drep-2* and heterozygous *dfmr1* single mutants were deficient in olfactory conditioning, but *drep-2$^{ex13}$/drep-2$^{ex13}$*; *dfmr1$^{B55}$/+* double mutants showed normal performance (*Figure 6E*). Thus, Drep-2 might be required downstream of mGluR signaling, probably in the context of synaptic translation, counteracting translational repression executed by FMRP. We observed that only chronic activation of mGluRs starting during development could improve the learning scores of *drep-2* mutants (*Figure 6C*). This result could be explained by chronic misregulation of neuronal translation in the mutants, rendering the synapses insensitive to enhanced mGluR signaling later in life.

At first glance, it might appear surprising that *drep-2$^{ex13}$/drep-2$^{ex13}$* and *dfmr1$^{B55/+}$* mutants cancel each other out. However, evidence for a tightly balanced control over synaptic translation has been provided: mutations in the gene *tsc2* cause tuberous sclerosis, a disease phenotypically similar to fragile X syndrome (FXS). Impaired long-term depression (LTD) in *tsc2*-mutant mice could be rescued by the application of mGluR agonists (*Auerbach et al., 2011*). Moreover, murine *fmr1* mutants showed exaggerated LTD, while the double mutant exhibited normal LTD. This demonstrated that synaptic proteins, if misregulated by either impaired or excessive mGluR-induced translation, impede appropriate LTD. In this manner, misregulation of opposing effectors can cause similar phenotypes, as is the case for the olfactory learning performance of *drep-2*, *dmGluRA*, and *dfmr1* mutants.

Fragile X associated tremor/ataxia syndrome (FXTAS) is a late-onset neurodegenerative disorder occurring in carriers of fragile X premutation repeats which is distinct from FXS. The presence of fragile X rCGG premutation repeats in flies activates the microRNA miR-277, which causes neurodegeneration (*Tan et al., 2012*). One of the targets negatively regulated by miR-277 is Drep-2. Notably, a putative *drep-2* hypomorph was shown to enhance the FXTAS neurodegenerative phenotype. While Drep-2 function per se was not investigated in this study and the mechanistic details remain rather elusive, this independent connection between Drep-2 and a scenario related to FXS is also suggestive.

Additional experiments will be required to both examine whether Drep-2 plays a role during translational regulation and to further explore its relationship to metabotropic signaling. Nevertheless, based on our findings, a function of Drep-2 in regulating mGluR-mediated local translation is a possible

scenario, which is now open for further investigation. In mammals, Dff-related CIDE proteins have, up to now, mainly been studied in fat tissue (*Yonezawa et al., 2011*). However, CIDEc is highly expressed in the mammalian brain (*Li et al., 2009*). It is now an intriguing possibility that Dff family proteins might play non-apoptotic neuronal and synaptic roles in mammals as well.

# Materials and methods

## Animal rearing and fly strains

All fly strains were reared under standard laboratory conditions (*Sigrist et al., 2003*) at 25°C and 65–70% humidity, with a constant 12/12 hr light/dark regimen. Flies were fed standard semi-defined cornmeal/molasses medium. Bloomington stock collection strain #5905, $w^{1118}$, was used as background for both the generation of transgenic animals (Bestgene, Inc., Chino Hills, CA) and for behavioral assays.

The following fly stocks were used: 30y-Gal4 (*Yang et al., 1995*), c305a-Gal4 (*Krashes et al., 2007*), elav$^{c155}$-Gal4 (*Lin and Goodman, 1994*), elav$^{L3}$-Gal4 (elav$^{III}$-Gal4) (*Luo et al., 1994*), gh146-Gal4 (*Stocker et al., 1997*), mb247-Gal4 (*Zars et al., 2000*), mb247::Dα7$^{GFP}$ (*Kremer et al., 2010*), UAS-Dα7$^{GFP}$ (*Leiss et al., 2009b*), Df$^{fw45−30n}$ (Bloomington stock #4966), dmGluRA$^{112b}$ and its control dmGluRA$^{2b}$ (*Bogdanik et al., 2004*), fmr1$^{B55}$ (*Inoue et al., 2002*), and $w^{1118}$ (*Hazelrigg et al., 1984*). All flies used for behavioral experiments were outcrossed to $w^{1118}$ for more than five generations in order to generate an isogenic genetic background.

## Generation of transgenic flies

Drep-2$^{ex13}$ mutants were generated using FLP–FRT recombination between the two stocks drep-2$^{d00223}$ and drep-2$^{e04659}$, as previously described (*Parks et al., 2004*). Drep-2$^{ex27}$ mutants were created in an analogous manner, using the transposon lines drep-2$^{e02920}$ and drep-2$^{e04659}$. In short, one of the elements containing a FRT site was combined with a line expressing the FLP recombinase under a heat shock promoter. These flies were crossed with the strain containing the second element to place both FRT sites *in trans*. Expression of FLP recombinase was triggered by a heat shock to 37°C. Offspring were collected and mutant candidates were validated by genomic PCR (forward primer: 5′-GCT GCT TGA GTA TGG GTG CA-3′; reverse primer: 5′-GGA GAC ATC CTC TCA AAG C-3′).

We generated transgenic flies expressing either plain drep-2 cDNA or eGFP- or mStrawberry-tagged drep-2 constructs, all under the UAS enhancer. The drep-2 cDNA LD32009 was amplified using the forward primer 5′-CAT GCC ATG GCA ATG GCC AGA GAG GAG TCT CGC-3′ and the reverse primer 5′-CGG GGT ACC AAT CT GTC CTC CTC ATC CTC TTC C-3′. The amplicon was inserted into the pEnter vector using NcoI and KpnI restriction sites. Invitrogen gateway cloning was used to create the expression constructs from pEnter. The vectors pTWG and pTGW (Carnegie Institution of Washington, Washington, DC) were used for generation of eGFP constructs; eGFP was replaced by mStrawberry by PCR for mStrawberry constructs.

Lack of dmGluRA in drep-2$^{ex13}$; dmGluRA$^{112b}$ double mutants was validated by single-fly PCR. The sequence of the primers for this PCR was as follows: forward primer: 5′-GGT GCC CCT TGC GGA CCA AA-3′; reverse primer: 5′- TTG TCG TCT GCG GCA CTG GG-3′. Lack of drep-2 was confirmed by stainings.

## Adult life span

In order to assay the life span, male flies were placed in groups of 25 animals in small food vials and transferred to fresh vials at least twice a week. Flies were kept at standard conditions. After each transfer, the number of dead and remaining live flies was counted. The number of days for each vial was determined at which 50% of flies were dead.

## Aversive olfactory conditioning

All experiments were conducted with three- to 5-day-old animals and carried out in a $w^{1118}$ genetic background. Flies were raised at 24°C and 60% relative humidity with a 14/10 hr light–dark cycle on cornmeal-based food prepared according to the Würzburg recipe (*Guo et al., 1996*). Flies were transferred to fresh food vials for up to 48 hr before behavioral experiments. Behavioral experiments were performed in dim red light at 70% relative humidity with 3-octanol (1:150 dilution in mineral oil presented in a 14 mm cup) and 4-methyl-cyclohexanol (1:100 dilution in mineral oil presented in a 14 mm cup) serving as olfactory cues and 120V AC current serving as a behavioral reinforcer.

Associative training was carried out following the single-cycle training procedure previously described (*Tully and Quinn, 1985*). Electric foot shock was applied after 10 s of odor presentation; afterwards twelve shock/odor pairings were conducted within 50 s. Odors and electric shock were applied in the same manner during conditioning as when testing for sensory acuity. STM was tested immediately after the end of the training session, 3 min after the onset of training. Performance of ITM and ARM was determined 3 hr after training; flies were transferred to neutral containers without food for the resting period. Two groups of flies were separately trained for separation of consolidated ARM and labile ASM, and one group was cooled in an ice-bath (0°C) for 90 s, 2.5 hr after training. Odor memory of this group was tested after a 30-min recovery period, that is, 3 hr after onset of training. Since labile ASM is erased by this procedure, performance of the cooled group is solely due to ARM.

Pharmaceutical components (MPEP (ab120008, Abcam, Cambridge, MA) and 1S,3R-ACPD (#0284, Tocris Bioscience, Bristol, United Kingdom)) were supplemented to liquid fly food, as previously described (*McBride et al., 2005*; *Tauber et al., 2011*). 1S,3R-ACPD was used in a concentration of 72.2 µM and MPEP at 9.7 µM (*Parmentier et al., 1996*; *McBride et al., 2005*, *2010*). Both compounds were diluted in $H_2O$. The same amount of $H_2O$ lacking any additional compounds was added to the control food. Flies were either raised throughout their entire development and adulthood on this food or, where indicated, were raised on control food lacking ACPD and only transferred to food containing ACPD after eclosion.

Calculation of behavioral indices was carried out as previously published (*Tully and Quinn, 1985*). ASM can be calculated by subtracting the performance of the cooled group from an uncooled group. Non-parametric tests (Mann–Whitney U test or Kruskal–Wallis) were used because of the small sample sizes. The significance level α was set to 5%. Asterisks are used to indicate significance in figures (* = $p < 0.05$; ** = $p < 0.01$; *** = $p < 0.001$; ns = $p \geq 0.05$). If several genotypes were compared, α and * symbols were adjusted by dividing the significance level by the number of comparisons (Bonferroni correction). Experimental data were analyzed using Microsoft Office 2011 and OriginLab (Northampton, MA) Origin Pro 9.0. Graphs were created using Gnuplot v4.6 (http://www.gnuplot.info) and Adobe Illustrator CS4.

## In situ hybridization

In situ hybridizations of whole mount embryos were performed as described by the Berkeley *Drosophila* Genome Project (http://www.fruitfly.org). The plasmid *LD32009* was cut using *Bam*HI and in vitro transcribed using Sp6 RNA polymerase to prepare antisense RNA probes. The plasmid was cut with *Sma*I and transcribed with T7 RNA polymerase to prepare sense probes.

## Antibodies

The *drep-2* cDNA LD32009 was amplified using the forward primer 5′-GAC CGT CGA CGT GGG TGT GGG AGC TGT CCA-3′ and the reverse primer 5′-GAC CCT CGA GTG AAT TCT GTC CTC CTC ATC CTC-3′. The amplicon was inserted into the pENTR4 vector (Invitrogen, Life Technologies, Carlsbad, CA) using *Sal*I and *Xho*I restriction sites. Invitrogen gateway cloning was used to create a 6xHis-tagged construct in pDEST17 (Invitrogen). A rabbit serum against this 6xHis-tagged C-terminal Drep-2 fusion protein (amino acids 252-483 of Drep-2-PA) was produced (Seqlab, Göttingen, Germany) and affinity-purified with the same fusion protein.

Antibody concentrations were as follows: mouse anti-Brp[Nc82] (*Wagh et al., 2006*) 1:100, guinea pig anti-Brp[N-Term] 1:800, mouse anti-ChAT[4B1] (*Yasuyama and Salvaterra, 1999*) 1:1000, rat anti-Dα7 (*Fayyazuddin et al., 2006*) 1:2000, mouse anti-Dlg[4F3] (*Parnas et al., 2001*) 1:500, mouse anti-DmGluRA[7G11] (*Panneels et al., 2003*) 1:100, rabbit anti-Drep-2[C-Term] 1:500, mouse anti-Fasciclin-II[1D4] (*Lin and Goodman, 1994*) 1:50, mouse anti-FMRP[5A11] (*Okamura et al., 2004*) 1:100, mouse anti-GFP[3E6] (Molecular Probes, Life Technologies) 1:500, rabbit anti-GFP (A11122, Life Technologies) 1:1000, guinea pig anti-Homer (*Diagana et al., 2002*) 1:200, rabbit anti-Syd-1 (*Owald et al., 2010*) 1:500, mouse anti-α-Tubulin[DM1A] (Sigma-Aldrich, St. Louis, MO) 1:100000, goat anti-mouse Alexa 488 (A11001, Invitrogen) 1:500, goat anti-rabbit Cy3 (111-167-003, Dianova, Hamburg, Germany) 1:500, goat anti-guinea pig Cy3 (106-166-003, Dianova) 1:500, donkey anti-rat Cy3 (712-165-153, Dianova) 1:250, goat anti-rabbit Atto 647N (40839, Sigma-Aldrich) 1:200, and goat anti-rabbit HRP (111-035-144, Dianova) 1:5000.

## Immunohistochemistry and imaging

Adult brains were dissected in ice-cold hemolymph-like saline (HL3) solution, fixed for 20 min in 4% paraformaldehyde (PFA) in 1x phosphate-buffered saline (PBS), pH 7.2, and then blocked in 5% normal

goat serum (NGS) in PBS with 0.3% Triton X-100 (PBT) for 30 min. The brains were incubated with primary antibodies together with 5% NGS for 48 hr at room temperature (RT) and then washed in PBT for 3 hr, followed by overnight incubation with secondary antibodies at RT. The brains were then washed for 3 hr with PBT and mounted in VectaShield (Vector Laboratories, Burlingame, CA) on slides. 3- to 7-day-old female flies were used for dissections.

Conventional confocal images were acquired at 21°C with a Leica Microsystems (Wetzlar, Germany) TCS SP5 confocal microscope using a 63×, 1.4 NA oil objective for detailed scans and a 20×, 0.7 NA oil objective for overview scans. Lateral pixel size was set to values around 90 nm for detailed scans. Exact values varied, depending on the situation. Typically, 1024 × 1024 images were scanned at 100 Hz using 4× line averaging. All images were acquired using the Leica LCS AF software.

Confocal stacks were processed using ImageJ software (http://rsbweb.nih.gov/ij). Deconvolution of images was conducted using MediaCybernetics (Rockville, MD) AutoQuant X2.1.1. Contrast was adapted for visualization, where necessary, using the levels tool in Adobe Photoshop CS4. Images shown in a comparison or quantified were processed with exactly the same parameters. Images were not post-processed before quantification, but exclusively afterwards and only for visualization. Cell body and active zone counts were quantified similarly as described previously (*Kremer et al., 2010*; *Christiansen et al., 2011*); the area of interest was segmented in ImageJ and then analyzed in Bitplane (Zürich, Switzerland) Imaris v6.23 using the surface tool. The cell body counts are comparable to the published number of 700 KCs in mb247-Gal4 (*Schwaerzel et al., 2002*). Active zone numbers were assessed via an anti-Syd-1 staining; counts were similar to the published number of 28,000–30,000 synapses in the calyx (*Kremer et al., 2010*).

STED microscopy was performed using a Leica Microsystems TCS STED setup equipped with a 100×, 1.4 NA oil immersion STED objective, as previously described (*Waites et al., 2011*). The depletion laser (Mai Tai Ti:Sapphire; Spectra Physics, Newport, Santa Clara, CA) was set to 760 nm. 1024 × 1024 STED images were scanned at 10 Hz using 2× line averaging. STED images were processed using linear deconvolution software integrated into the Imspector software (Max-Planck-Innovation, München, Germany).

## Immunoelectron and electron microscopy

Brains were dissected in HL3 solution and fixed for 20 min at RT with 4% paraformaldehyde and 0.2% glutaraldehyde in a buffer containing 50 mM sodium cacodylate and 50 mM NaCl at pH 7.5. Afterwards, brains were washed twice in the buffer and dehydrated through a series of increasing alcohol concentrations. Samples were embedded in LR-Gold resin by incubation in ethanol/LR-Gold 1:1 solution overnight at 4°C, followed by ethanol/LR-Gold 1:5 solution for 4 hr at RT and, finally, 3× with LR-Gold/0.2% benzil once overnight, then for 4 hr and again overnight. Thereafter, the brains were placed in BEEM (West Chester, PA) capsules covered with LR-Gold/0.2% benzil resin and placed under a UV lamp at 4°C for 5 days to allow for polymerization of the resin.

Following embedding, 70–80 nm sections were cut using a Leica Ultracut E ultramicrotome equipped with a 2-mm diamond knife. Sections were collected on 100 mesh nickel grids (Plano, Wetzlar, Germany) coated with 0.1% Pioloform resin and transferred to a buffer solution (20 mM Tris–HCl, 0.9% NaCl, pH 8.0). Prior to staining, sections were blocked for 10 min with 0.04% BSA in buffer. Sections were incubated with the primary antibody in blocking solution overnight at 4°C. After washing 4× in buffer, sections were incubated in buffer containing the secondary antibody (goat anti-rabbit 10 nm colloidal gold, BBI Solutions, Cardiff, United Kingdom, 1:100) for 2–3 hr at RT. Finally, the sections were washed 4× in buffer and 3x in distilled water. Contrast was enhanced by placing the grids in 2% uranyl acetate for 30 min, followed by 3× washing with water and, afterwards, incubation in lead citrate for 2 min. The grids were then washed 3× with water and dried. Images were acquired on a FEI (Hillsboro, OR) Tecnai Spirit, 120 kV transmission electron microscope equipped with a FEI 2K Eagle CCD camera.

## Western blot analysis of adult heads

Fly head protein extraction was performed as follows: flies were decapitated and 20 heads of each genotype were sheared manually in 40 µl of 2% SDS aqueous solution using a micropistil fitting tightly into a 1.5-ml cup. An amount of 4 µl of a 10% Triton-X 100 aqueous solution and 40 µl of 2× sample buffer (*Laemmli, 1970*) was added, and samples were heated at 95°C for 10 min. After centrifugation for 5 min at 16,000×g, in order to pellet the debris, 8.4 µl of the sample (equivalent to two fly heads) was subjected to denaturing SDS-PAGE using an 8% Tris-Cl gel. Proteins were transferred onto a

nitrocellulose membrane, blocked with 5% skim-milk in 1× PBS supplemented with 0.1% Tween-20 and probed with affinity-purified rabbit anti-Drep-2[C-Term] (#7183; 1:5000) diluted in 5% skim-milk in 1x PBS, supplemented with 0.1% Tween-20, followed by washing steps. Secondary anti-rabbit IgG horseradish peroxidase (HRP)-conjugated antibodies (Dianova) and an enhanced chemoluminescence detection system (RPN 2232, GE Healthcare (Little Chalfont, United Kingdom) ECL Prime) with Hyperfilm ECL (GE Healthcare) were used for detection. After $NaN_3$ treatment, membranes were reprobed for α-Tubulin as a loading control, using the monoclonal antibody DM1A (Sigma; 1:100000).

## Head fractionation and enrichment blot (synaptosome-like preparation)

The *Drosophila* head fractionation protocol has recently been published (*Depner et al., 2014*). It is based on protocols from mammalian subcellular preparations (*Huttner et al., 1983*; *Ahmed et al., 2013*). In brief, *Drosophila* wild-type heads were sheared mechanically in the absence of detergents and differential centrifugation was applied to separate particles according to their size and density. Analytical samples from the fractions obtained were taken and the protein concentration determined. An amount of 10 µg total protein from each fraction was subjected to SDS-PAGE (10% polyacrylamide Tris-Tricine gel (*Schägger, 2006*)), followed by immunoblotting for Drep-2[C-Term]. HRP-conjugated goat anti-rabbit antibodies (111-035-144, Dianova, 1:5000) were used for ECL detection. Signals were recorded using an ImageQuant LAS 4000 image reader (GE Healthcare). Films were scanned in transmission mode (Epson (Long Beach, CA) V770 scanner).

## Nuclease activity assay

The drep-2 cDNA *LD32009* was fused to an N-terminal His-tagged maltose-binding protein for expression and purification of Drep-2. Drep-2 was transformed in *Escherichia coli* BL21 Rosetta2 (DE3) pLys cells (NEB, Ipswich, MA). Drep-2 was cultured in TB medium at 37°C until an OD of ~1.0 was reached and subsequently cooled down to 20°C. Protein expression was induced by the addition of 0.5 mM IPTG. Cells grew overnight and were harvested by centrifugation (6 min, 6000 rpm at 4°C). The Drep-2 pellet was resuspended in 20 mM Tris/HCl pH 7.4, 250 mM NaCl, 8 mM imidazole, and 1 mM DTT. Cells were lysed by sonication at 4°C and the supernatant was cleared by 45 min centrifugation (21,500 rpm at 4°C). A $Ni^{2+}$-NTA (cv ~1 ml; GE Healthcare) was equilibrated with 20 mM Tris/HCl pH 7.4, 250 mM NaCl, 1 mM $MgCl_2$, 8 mM imidazole, and 1 mM DTT.

Drep-2 was loaded onto the column and washed with 3 cv of equilibration buffer. Drep-2 was eluted in a linear gradient to 20 mM Tris/HCl pH 7.5, 250 mM NaCl, 400 mM imidazole, and 1 mM DTT. The maltose-binding protein was cleaved by TEV proteases, yielding untagged Drep-2 protein, during dialysis in 20 mM Tris/HCl pH 7.4, 100 mM NaCl, and 1 mM DTT and loaded onto a MonoQ 10/100 column (GE Healthcare) equilibrated with 20 mM Tris/HCl pH 7.4 and 1 mM DTT. Drep-2 was eluted in a linear gradient from 100 to 1000 mM NaCl. Size exclusion chromatography was performed with a HighLoad Superdex S200 16/60 column (GE Healthcare) equilibrated with 20 mM Tris/HCl pH 7.5, 250 mM NaCl, and 1 mM DTT.

The actual nuclease activity assay was conducted as follows: an amount of 10 µg of Drep-2 was incubated with 0.3 µg linearized *pUC19* plasmid DNA in 15-µl reaction buffer (20 mM Hepes-NaOH pH 7.4, 50 mM NaCl, 5 mM $MgCl_2$) at 37°C in limited digestion experiments. Aliquots were taken at different time intervals and the reaction was stopped by DNA loading dye containing 10 mM EDTA. Samples were electrophoretically separated over a 1% (wt/vol) agarose gel containing ethidium bromide.

## Quantitative affinity purification and mass spectrometry

Drep-2 in vivo interaction partners were identified using affinity purification and mass spectrometry (q-AP-MS; *Figure 7A*). The major challenge in such experiments is to distinguish true interaction partners from non-specific contaminants. Q-AP-MS can solve this problem by comparing the abundance of identified proteins with a control (*Vermeulen et al., 2008*; *Paul et al., 2011*).

Drep-2[GFP] was expressed using elav[c155]-Gal4. We purified Drep-2[GFP] from fly heads using a single chain anti-GFP antibody coupled to agarose beads. We performed parallel pulldowns on the same lysates using control agarose beads to control for unspecific binding. As an additional control, GFP-negative lysate from wild-type flies was included into the experiment. We identified a total of 3284 proteins in the pulldown experiments. 202 proteins were significantly enriched in GFP pulldowns compared to plain bead controls. 35 of these proteins could be confirmed in comparative analysis with GFP pulldowns of wild-type lysates and were, therefore, defined as robust interactors/core proteins (permutation-based FDR = 1%; $S_0$ = 1, *Figure 7B*).

Pulldown experiments for the quantitative affinity purification were conducted in the following manner: an amount of 500 µl of fly heads was immersed in liquid nitrogen and pulverized mechanically with a BioPulverizer (Biospec Products, Bartlesville, OK). Powdered tissue was homogenized in 500 µl cold lysis buffer (50 mM Tris–HCl (pH 7.6), 150 mM NaCl, 1 mM MgCl$_2$, 1 mM EDTA, 10% glycerol, 0.4% DOC, and protease inhibitors (Complete Mini, Roche Diagnostics, Indianapolis, IN)). After incubation on ice for 30 min, 500 µl lysis buffer without DOC and Triton were added to a final concentration of 1%. Samples were centrifuged at 14,000×g for 15 min at 4°C to remove insoluble material. The supernatant was transferred to a fresh tube for pull-down experiments.

Immunoprecipitations of GFP-tagged bait proteins were performed in triplicate using GFP-Trap agarose beads (Chromotek, Planegg-Martinsried, Germany) according to the manufacturer's instructions. Soluble protein fractions were incubated with either 25 µl of GFP-Trap or plain control beads for 60 min at 4°C under constant rotation. The beads were washed twice with washing buffer (50 mM Tris–HCl (pH 7.6), 150 mM NaCl, 1 mM MgCl$_2$, 1 mM EDTA, 10% glycerol) and once with PBS. Proteins bound to the beads were eluted by applying 50 µl elution buffer (6 M urea/2 M thiourea) twice and proceeded to in-solution digestion followed by LC-MS/MS analysis.

Liquid chromatography MS/MS analysis was performed as follows: protein eluates were reduced for 30 min at RT in 10 mM dithiothreitol solution, followed by alkylation by 55 mM iodacetamide for 20 min in the dark at RT. The endoproteinase LysC (Wako, Osaka, Japan) was added following a protein:enzyme ratio of 50:1 and incubated for 4 hr at RT. After dilution of the sample with 4x digestion buffer (50 mM ammonium bi-carbonate in water (pH 8.0)), sequence grade modified trypsin (Promega, Madison, WI) was added (same protein:enzyme ratio as for LysC) and digested overnight. Finally, trypsin and Lys-C activity was quenched by acidification of the reaction mixtures with TFA to pH ~2. Afterwards, peptides were extracted and desalted using StageTips (*Rappsilber et al., 2003*).

Peptide mixtures were separated by reversed phase chromatography using the EASY-nLC system (Thermo Scientific, Waltham, MA) on in-house manufactured 20 cm fritless silica microcolumns with an inner diameter of 75 µm. Columns were packed with ReproSil-Pur C18-AQ 3 µm resin (Dr. Maisch GmbH, Ammerbuch-Entringen, Germany). Peptides were separated on an 8–60% acetonitrile gradient (214 min) with 0.5% formic acid at a nanoflow rate of 200 nl/min. Eluting peptides were directly ionized by electrospray ionization and transferred into a Q Exactive mass spectrometer (Thermo Scientific). Mass spectrometry was performed in the data-dependent positive mode with one full scan (m/z range = 300-1700; R = 70,000; target value: 3 × 106; maximum injection time = 120 ms). The ten most intense ions with a charge state greater than one were selected (R = 35,000, target value = 5 x 105; isolation window = 4 m/z; maximum injection time = 120 ms). Dynamic exclusion for selected precursor ions was set to 30 s.

MS/MS data were analyzed by MaxQuant software v1.2.2.5 as described (*Cox et al., 2011*). The internal Andromeda search engine was used to search MS/MS spectra against a decoy *D. melanogaster* UniProt database (DROME.2016-06) containing forward and reverse sequences. The search included variable modifications of methionine oxidation and N-terminal acetylation, and fixed modification of carbamidomethyl cysteine. Minimal peptide length was set to six amino acids and a maximum of two missed cleavages was allowed. The FDR was set to 0.01 for peptide and protein identifications. If the identified peptide sequence set of one protein was equal to or contained another protein's peptide set, these two proteins were grouped together and the proteins were not counted as independent hits.

Label-free quantification (LFQ) was performed in MaxQuant as described (*Hubner et al., 2010*). Unique and razor peptides were considered for quantification with a minimum ratio count of 1. Retention times were recalibrated based on the built-in nonlinear time-rescaling algorithm. MS/MS identifications were transferred between LC-MS/MS runs with the 'Match between runs' option, in which the maximal retention time window was set to 2 min. For every peptide, corresponding total signals from multiple runs were compared to determine peptide ratios. Median values of all peptide ratios of one protein then represent a robust estimate of the protein ratio. LFQ intensity values were logarithmized and missing values were imputed with random numbers from a normal distribution whose mean and standard deviation were chosen to best simulate low abundance values below the noise level (width = 0.3; shift = 1.8). GFP pull-down samples and plain-bead control samples were selected as individual groups of three technical replicates each; significantly enriched proteins were determined by a volcano plot-based strategy, combining standard two-sample t-test p-values with ratio information. Significance corresponding to an FDR of 1, 5, or 10% was determined by a permutation-based method (*Tusher et al., 2001*).

The network of biochemical interactions (*Figure 7D*) was created using Microsoft Excel 2011, Cytoscape v2.8.3/v3.0.0 (http://www.cytoscape.org) and Adobe Illustrator CS4, following the protocol available at http://protocols.andlauer.net/cytoscape.pdf.

## Pulldown experiments

Flies of the genotype *elav^c155-Gal4; uas-drep-2^GFP* were used for preparations containing Drep-2[GFP] and flies of the genotype *elav^c155-Gal4; uas-syd-1^GFP* (*Owald et al., 2010*) were used for preparations containing Syd-1[GFP]. For each experiment, 500 µl of adult fly heads were mechanically homogenized in 500 µL lysis buffer (50 mM Tris pH 8.0, 150 mM KCl, 1 mM MgCl2, 1 mM EGTA, 10% glycerol-containing protease inhibitor cocktail (Roche Diagnostics)). Sodium deoxycholate (10%) was added to achieve a final concentration of sodium deoxycholate of 0.4% and the lysate was incubated for 30 min on ice. The lysate was diluted 1:1 with sodium deoxycholate-free lysis buffer, then 10% Triton X-100 was added for a final concentration of 1% Triton X-100, and the lysate was rotated at 4°C for 30 min.

After centrifugation for 15 min at 16,000×g, the supernatant was used in immunoprecipitations with GFP-Trap-A beads and blocked agarose beads as binding control (Chromotek). After incubation at 4°C overnight, the beads were washed in buffer without detergent and glycerol. Proteins were eluted from the beads with SDS sample buffer. The samples were separated by one-dimensional SDS-PAGE gradient gel (TGX 4–12% precast, Bio-Rad, Hercules, CA).

Proteins were transferred onto a nitrocellulose membrane and probed with mouse anti-FMRP[5A11] (1:100). A secondary anti-mouse IgG horseradish peroxidase (HRP)–conjugated antibody (Dianova) and an enhanced chemoluminescence (ECL) detection system with Hyperfilm ECL (GE Healthcare) were used for detection. After NaN$_3$ treatment, the membranes were re-probed with rabbit anti-Drep2[C-term] (1:2000) and rabbit anti-GFP (1:1000) (Life technologies, A11122). Films were scanned in transmission mode (Epson V770) and images were imported to Adobe Photoshop.

## Acknowledgements

We thank Madeleine Brünner, Anastasia Stawrakakis and Nicole Holton for excellent technical assistance. We thank Reinhard Jahn for the opportunity to perform experiments in his lab at the Max Planck Institute for Biophysical Chemistry in Göttingen, Germany. We are grateful to Irmgard Sinning, Mikiko Siomi, Uli Thomas, and Hugo Bellen for generously contributing antibodies. This work was supported by the following grants: Deutsche Forschungsgemeinschaft (DFG) to SJS: SFB958/A6, Exc257, FOR1363; DFG to SSK and MSc: SCHW1410/1-1; DFG to BL and MCW: SFB958/A6; DFG to AF and SD: SFB 889/B9. Freie Universität Berlin DynAge Focus Area to SJS. German Ministry of Research and Education (BMBF) via the Bernstein Center Göttingen (BCCN II/B1) to AF and SD: grant number 01GQ1005A.

## Additional information

### Funding

| Funder | Grant reference number | Author |
|---|---|---|
| Deutsche Forschungsgemeinschaft | Exc257, FOR1363 | Stephan J Sigrist |
| Freie Universität Berlin | DynAge Focus Area | Stephan J Sigrist |
| Deutsche Forschungsgemeinschaft | SCHW1410/1-1 | Sabrina Scholz-Kornehl, Martin Schwärzel |
| Deutsche Forschungsgemeinschaft | SFB 889/B09 | Shubham Dipt, André Fiala |
| Bundesministerium für Bildung und Forschung via the Bernstein Center Göttingen (BCCN II/B1) | 01GQ1005A | Shubham Dipt, André Fiala |
| Deutsche Forschungsgemeinschaft | SFB958/A6 | Bernhard Loll, Markus C Wahl, Stephan J Sigrist |

The funders had no role in study design, data collection and interpretation, or the decision to submit the work for publication.

## Author contributions

TFMA, Conception and design, Acquisition of data, Analysis and interpretation of data, Drafting or revising the article; SSK, Acquisition of data, Analysis and interpretation of data; MK, HD, Acquisition of data, Analysis and interpretation of data, Drafting or revising the article; SD, Acquisition of data, Analysis and interpretation of data; RT, HAB, CQ, VKG, Acquisition of data, Drafting or revising the article; MGH, Conception and design, Analysis and interpretation of data; MC, Acquisition of data, Drafting or revising the article; BL, Conception and design, Acquisition of data, Analysis and interpretation of data, Drafting or revising the article; MCW, AF, MSe, Conception and design, Drafting or revising the article; MSc, Conception and design, Analysis and interpretation of data; SJS, Conception and design, Analysis and interpretation of data, Drafting or revising the article

## Author ORCIDs

Till F M Andlauer, http://orcid.org/0000-0002-2917-5889

## Additional files

### Supplementary files

• Supplementary file 1. Mass spectrometry: core proteins enriched over both controls. The table shows proteins enriched at an FDR of 1%. Ranks are based on the ratios of Drep-2$^{GFP}$ IPs, GFP beads vs. plain beads (compare *Figure 7A–B*). Putative functions were derived from Flybase. GFP (#6) was removed from this list.

• Supplementary file 2. Mass spectrometry: all proteins enriched at an FDR of 10%. The table shows two ratios for all proteins: first, Drep-2$^{GFP}$ animals, GFP beads vs. plain beads and, second, GFP beads, Drep-2$^{GFP}$ vs. wild-type (wt) animals (compare *Figure 7A*). "+" symbols indicate whether a protein was significantly enriched at a given FDR. Proteins are sorted according to the first ratio. Proteins were labeled in cyan if significantly enriched for both ratios at an FDR of 1%; these proteins constitute the core interactors. Protein names were labeled in red if the protein was positively enriched for the first ratio but negatively enriched for the second one, and thus probably constitutes a false positive. Protein names were labeled in shades of green if the protein was included as an additional interactor in the network of interacting proteins (*Figure 7D*): dark green = FDR 1%, medium green = FDR 5%, light green = FDR 10%.

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
