## [Decision Letter]

Thank you for sending your work entitled “Drep-2 is a novel synaptic protein important for learning and memory” for consideration at *eLife.* Your article has been favorably evaluated by K VijayRaghavan (Senior editor) and 4 reviewers, one of whom is a member of our Board of Reviewing Editors.

The Reviewing editor and the other reviewers discussed their comments before we reached this decision, and the Reviewing editor has assembled the following comments to help you prepare a revised submission:

This paper presents a beautiful characterization of Drep2, a surprising new learning and memory mutant. The STED and co-localization experiments played well with the genetic interactions with dFMR1 and mGluRA in olfactory learning and take this out of the realm of simply being a “new gene” identification paper. However, after a long discussion amongst the reviewing editor and the three reviewers there was agreement that this version of the manuscript is not suitable for publication. The overall consensus was that the manuscript needs to be streamlined both in terms of the data that are presented and in terms of the interpretation of the experiments, especially the relationship of Drep2 to dFMR1 and mGluRA. While the central findings in the paper (localization, role in olfactory learning) are solid, the way the paper is structured, the numerous ancillary data sets and the unfocused discussion tend to distract from what could be a strong message. To some degree these might be considered cosmetic problems, but they make the study seem much weaker than it probably is and more importantly, the lack of clarity in the Discussion results in a somewhat misleading bottom line about Drep2 function.

I am appending the complete reviews for your consideration, but the main points that need to be addressed if you choose to submit a revision are enumerated below. The reviewers all feel that addressing these issues in the manuscript would make this a much stronger vehicle for highlighting the interesting central findings on Drep2 localization and its role in olfactory associative learning.

1) Ethanol data. This data set did not seem particularly “on point” for the main conclusions of the paper and only served to distract the readers from the take home message. More concerning was that there were a number of controls that were either not done or not reported, which makes the conclusions of this part of the study tenuous. Given that these data are not central to the manuscript's main point, recommend they be removed.

2) Courtship data. In the view of several of the reviewers, this data set is not interpretable as demonstrating a learning/memory defect. The fact that the mutants do not show any appreciable level of initial courtship means that there was no opportunity for these males to actually learn since performance of courtship is required for training (the behavior has operant features). While the reintroduction of Drep2 rescues learning, it also rescues initial courtship meaning this experiment does not necessarily imply that Drep2 has a role in courtship learning, it could have simply allowed the animals to be trained. We recommend these data are removed since they are not central to the paper.

3) Imaging data. The alpha lobe imaging data were felt to be peripheral to the point of the paper, not because this would not be a valid way of looking at the mutant, but rather that the particular experiment the authors chose to do was not particularly relevant. Why look at the alpha lobes? Changes in Ca+2 and its function in learning is better understood elsewhere like the gamma lobes, and given the clear role of Drep2 in calyx why not look there? The alpha lobes are far downstream from the calyx and the presumptive site of Drep2 action, and probably not important for STM. It is not at all clear what the difference shown represents. The data as they stand could be removed, though the paper could conceivably be strengthened by addition of imaging from either gamma lobe or calyx.

4) Statistics. The authors should explain and justify the particular non-parametric statistical tests they used- are the data not normally distributed?

5) Writing. Language was in many places rather stilted and could used editing to make sure that it is clear. More problematic was the repetitious nature of much of the Introduction and the over-written Discussion that never really gave the reader a clear take-home message (see point 6 below).

6) Mechanism/model for Drep2 function. While this paper suggests that Drep2 is acting in some way with mGluRA and dFMR1, the exact mechanism is not clear. The data strongly suggest that this interaction may happen early in development (or at least have an important developmental component) since rescue only works if the protein is present developmentally. The authors are encouraged to show the negative result with the adult-only rescue since this constrains models. While the current data make it impossible to posit a very specific model, the authors should at least give a cogent set of possibilities. As is, there is an implication in the paper that Drep2 may function like mGluR acutely and this is misleading.

Reviewer #1:

In this manuscript the authors present a very extensive characterization of mutants in Drep-2, a protein initially thought to be involved in apoptosis, which now they convincingly show to be associated with synapses and at least in the adult mushroom bodies resides on the postsynaptic side. The authors convincingly show that the protein has a role in glutamatergic signaling-dependent olfactory learning and memory.

Nevertheless there are multiple issues to be resolved to improve presentation and above all readability of the manuscript!

1) The manuscript is way too long, unnecessarily in my opinion.

2) The writing style and language need to be improved significantly.

3) There is a plethora of data in the supplement that do not add much to the conclusions of the already very large data set.

4) The Discussion is unnecessarily too long in my opinion.

More specifically:

1) The language and style must be improved because as it is the manuscript does not read well, is full of errors in syntax, colloquialisms and sections appear disconnected as if written in different times by different people. There appears to be no rhyme or reason in the way the Materials and Methods section was assembled. Genetics and husbandry, transgenics, longevity and behavioral methods customarily are placed in proximity, whereas the molecular techniques are grouped together.

2) Figures 1 and 2 do not follow the text presentation and in my opinion would benefit the manuscript and help the reader if Figure 1 was made in sequence of panels: Figure 1, Figures 1 and 2
Figure 2 should then be the fractionation (current Figure 1) and Figure 1—figure supplement 1. The text in the results sections should be changed to reflect this.

3) What are the bands around 40kD in the fractionation blot? What are the different fractions? What do the names P1 slof and solid mean? Please describe.

4) Why is counting the number of Kenyon cells in control and mutant animals a measure of apoptosis?

5) Why use mushroom body driver c305a with its limited coverage of MB neurons rather than any other one including mb247 which the authors use for many other applications?

6) Although it is very nice data, how does inclusion of all the ethanol sensitivity data add to a paper dealing with learning and memory, at least according to its present title?

7) Shouldn't the calcium imaging data be part of the main figures of the manuscript. They present a significant point, but if part of the main manuscript the effect of the mutation must be rescued.

Reviewer #2:

This new manuscript from Andlauer et al. provides a functional description of a new postsynaptic protein, Drep-2, which genetically interacts with FMRP and mGluRA in associative learning. The findings are significant and interesting. The work is technically impressive. The authors move from a protein with a completely unknown function, through a series of impressive localization studies that demonstrate enrichment around post-synaptic densities. They generate null alleles, and then use these alleles demonstrate a role for Drep-2 in alleles in associative short and intermediate term memory. Expression of Drep-2 in the mushroom body neurons is sufficient to rescue the memory phenotypes of the Drep-2 null alleles. Importantly, the authors show that the Drep-2 short term memory phenotype is also rescued by developmental activation of mGluRA receptors and by mutation of the dFMR1, demonstrating a genetic interaction with the proteins. The authors also perform proteomics to isolate the Drep-2 core network of interacting proteins, which include FRMP and several proteins involved in translational control. This work takes Drep-2 from an unknown function to a new component in the FMRP pathway regulating postsynaptic plasticity.

Interestingly, the authors also present some important negative results: that contrary to naïve expectations, Drep-2 does not appear to affect apoptosis and that the loss of this protein does not impact synapse structure. Overall, I believe these experiments were carefully and thoughtfully performed.

The author's movement through the characterization of Drep-2 is impressive, but also somewhat uneven. The authors further examined the Drep-2 null alleles for phenotypes and genetics interactions in courtship conditioning, locomotor behavior and ethanol sedation behavioral paradigms to show demonstrate Drep-2 antagonizes dFRM1. The null allele has a phenotype in each of these behaviors. In courtship conditioning and locomotor behavior, the authors do not show that the phenotype is due to the loss of Drep-2 through genetic rescue. Also in courtship conditioning, an unpublished automated system is used to score the many assays they have run. The median learning scores using this system are quite low and may miss many courtship behaviors. Alternatively, the assay set up may be suppressing courtship. The authors also do not provide any indications if hyperactivity can provide false positives for courtship with this system. Is it possible that hyperactive drep-2EX13 flies are not courting, and therefore not learning, but their high level of activity is somehow scored as courting behavior in this system?

The ethanol sedation phenotype was rescued, but I had some concerns. In the description of this experiment, the authors focused on Drep-2 expression in the ring neurons of the ellipsoid body—the site of the homer requirement in ethanol sedation and tolerance. However, they never show that the Drep-2 is required in these neurons. When discussing the absence of an interaction with mGluRA, the authors should also cite the possibility that Drep-2 may be required outside of the ring neurons for normal levels of sedation. It is common to perform ethanol sedation experiments separately in males and female as the differences in body size lead to large differences in the rate of sedation. As far as I can tell, this was not done and there is no indication of whether the number of females/males was controlled between genotypes. The rate of sedation was so quick, that there may be a floor effect for the drep-2 mutants, which could confound the ability of ACPD to rescue this phenotype. Also, I am unclear of the status of the white gene in the mutant genotypes. The level of white activity can dramatically impact the rate of sedation (Chan, et al. 2014. Alcoholism: Clinical and Experimental Research 38:1582-1593). It is highly unlikely that the rescue is due to increasing the copies of mini-white, but the authors need to indicate whether the mutants are in a w1118 or w+ background. Also the authors need to justify using frozen fly weight and not protein content as was done in Moore, et al., to normalize ethanol content in the absorption studies. Although, these experiments on ethanol sedation are potentially interesting and could add value to this manuscript, they are not necessary.

Reviewer #3:

This manuscript describes a thorough set of experiments to characterize the role of the CIDE-N domain protein drep-2 in the Drosophila CNS. The role of this protein in regulating learning and memory, presumably via modulation of the postsynaptic terminal in the MB calyx, is novel, and the functional antagonism with dfmr intriguing. The experiments are a tour de force, and the major conclusions could open some promising avenues of future investigation (e.g., of fragile X phenotypes). I have no major concerns.

Minor comments:

This is a dense manuscript, and a schematic outlining the expression patterns of the various proteins (drep-2, mGluR, dfmr), as well as showing the hypothesized interactions between them would significantly help the reader to digest the conclusions. While the nature interaction between drep-2 and mGluR signaling is not completely worked out, showing the directionality of the effects of each manipulation on learning would help.

Figure S1 shows some nice imaging data suggesting that the MB responses to odorants are attenuated. In panel C, error bars are missing for the drep-2 trace during the odor presentation window (in my copy, at least). Also, the caption should note the location of the recordings (tip of the alpha lobe).

In the last paragraph of the Introduction, the authors abruptly break from discussing previous findings into a discussion of the conclusions in the current manuscript. The last 3-4 sentences are better suited to the Results and Discussion. The authors could replace them with something to the effect of “Here we have examined a potential synaptic role for Drep-2, a CIDE-N domain protein, and its role in regulating learning and memory through interaction with mGluR signaling”.

The authors state: “Thus, we obviously produced two null situations for drep-2 and proved the specificity...”. This sounds odd (the “proved” part); I suggest changing to something along the lines of “Thus, the drep-2C-Term antibody demonstrated specificity in both western blots and immune staining. Based on the lack of detectable expression, we conclude that both mutants are null alleles.”

Also: “The adenylyl cyclase Rutabaga mediates coincidence detection between the conditioned and the unconditioned stimulus in KC γ-lobes (Qin et al., 2012)...” Rut-dependent coincidence detection has been directly observed in the alpha lobe by two groups (Tomchik and Davis, 2009; Neuron 64: 510-521; Gervasi et al., 2010; 65: 516-529); these papers should be cited here, particularly since the alpha lobe was imaged.

Reviewer #4:

This paper provides a very thorough characterization of Drep2, a novel synaptic protein with a function in learning and other behaviors. The authors show convincing mutant phenotypes and provide a truly detailed and beautiful accounting of where Drep2 is expressed and its subsynaptic localization in Kenyon cells. This careful work and the novel nature of the Drep2 protein make the work interesting.

The main problem I have with this manuscript is the lack of actual molecular insight into Drep2 function. The data on dfmr and mGluRA are suggestive, but the story is clearly not straightforward and trying to make it fit the dfmr/mGluR template just does not work; it only serves to highlight the fact that something else is going on and the lack of concrete mechanism. There are several pieces of data that lead to significant questions that are not answered in this paper.

A) mGluRA is not present in the mass spec data set. This does not support a direct adult interaction.

B) The data showing normal mGluR levels in Drep2 mutants is also not consistent with direct interaction.

C) The fact that mGluR agonists can only influence the Drep2 phenotype if present during development is a huge red flag. If there was a direct interaction, one should be able to get acute interaction. This smacks of some either circuit-level or pathway-level compensation that could be very indirect.

D) There is no evidence that the interaction between mGluR and Drep2 occurs in KCs. The authors use drugs and this makes things very hard to interpret. They cite the Kanellopoulos paper which uses ACUTE mGluR RNAi to demonstrate interaction in KCs with dfmr. This cannot be extrapolated to Drep2 because of the developmental nature of that interaction. Again, the issue of circuit level compensation becomes important.

E) The courtship data do not necessarily suggest a “learning” interaction of dfmr and Drep2. Dfmr males do not court and if males do not court in this paradigm they cannot learn since there is an operant component to this behavior. Drep2 mutants do court and loss of Drep2 in the double rescues courtship. The fact that the dfmr;Drep2 males now court could be the basis of the rescue, not an enhancement of learning.

In the final analysis, while the involvement of a protein in the Diff family in synapses is interesting, without a novel and clear mechanism for its function this paper is not going to be of general interest.

---

## [Author Response]

*1) Ethanol data. This data set did not seem particularly “on point” for the main conclusions of the paper and only served to distract the readers from the take home message. More concerning was that there were a number of controls that were either not done or not reported, which makes the conclusions of this part of the study tenuous. Given that these data are not central to the manuscript's main point, recommend they be removed*.

We agree that the ethanol data did not really fit to the story of the manuscript. Therefore, we had originally placed it in supplemental figures only addressed in the Discussion. The data were included because we originally wanted to show all mutant phenotypes we had observed. However, we fully agree that there were too many ancillary datasets included in this manuscript. Thus, to improve the flow of the manuscript, we have now concentrated on olfactory conditioning and removed ethanol sensitivity, locomotor hyperactivity, courtship conditioning, and functional imaging experiments.

*2) Courtship data. In the view of several of the reviewers, this data set is not interpretable as demonstrating a learning/memory defect. The fact that the mutants do not show any appreciable level of initial courtship means that there was no opportunity for these males to actually learn since performance of courtship is required for training (the behavior has operant features). While the reintroduction of Drep2 rescues learning, it also rescues initial courtship meaning this experiment does not necessarily imply that Drep2 has a role in courtship learning, it could have simply allowed the animals to be trained. We recommend these data are removed since they are not central to the paper*.

We agree and the data have been removed.

*3) Imaging data. The alpha lobe imaging data were felt to be peripheral to the point of the paper, not because this would not be a valid way of looking at the mutant, but rather that the particular experiment the authors chose to do was not particularly relevant. Why look at the alpha lobes? Changes in Ca+2 and its function in learning is better understood elsewhere like the gamma lobes, and given the clear role of Drep2 in calyx why not look there? The alpha lobes are far downstream from the calyx and the presumptive site of Drep2 action, and probably not important for STM. It is not at all clear what the difference shown represents. The data as they stand could be removed, though the paper could conceivably be strengthened by addition of imaging from either gamma lobe or calyx*.

We have imaged the alpha lobes because of methodological reasons. However, we agree that these data are not well suited to draw conclusions regarding the function of Drep-2 in the calyx. We have therefore removed the data and now only refer to them briefly in the Discussion as preliminary data. We added the sentence: “Interestingly, preliminary experiments indicate that the learning deficits of drep-2 mutants might involve decreased Ca^2+^ responses in KCs (not shown).”

*4) Statistics. The authors should explain and justify the particular non-parametric statistical tests they used- are the data not normally distributed*?

In the current Figures 3 and 5, Figure 5—figure supplement 1, and Figure 6, several samples had sizes below 20, making it difficult to reliably assess whether the data are normally distributed or not. We thus preferred to use non-parametric tests (Kruskal-Wallis and Mann-Whitney U). Although some of our experiments had higher sample sizes, we did not want to mix different tests for statistical significance and decided to use non-parametric tests for all experiments.

We used additional median-based methods for the courtship conditioning and ethanol sensitivity data, both of which have now been removed. This was due to the incidence of a few extreme outliers that skewed the results when parametric/mean-based methods were used. Median-based assessment of the data had thus delivered results that better corresponded to the true distribution of data points.

*5) Writing. Language was in many places rather stilted and could used editing to make sure that it is clear. More problematic was the repetitious nature of much of the Introduction and the over-written Discussion that never really gave the reader a clear take-home message (see point 6 below)*.

We apologize for the poor language of our manuscript. We worked on the manuscript for a long time and the repeated addition and removal of sentences and sections obviously led to poor readability. We have now worked on both language and readability and have also involved native English speakers for language corrections. We really hope that the flow and language level have been substantially improved.

*6) Mechanism/model for Drep2 function. While this paper suggests that Drep2 is acting in some way with mGluRA and dFMR1, the exact mechanism is not clear. The data strongly suggest that this interaction may happen early in development (or at least have an important developmental component) since rescue only works if the protein is present developmentally. The authors are encouraged to show the negative result with the adult-only rescue since this constrains models. While the current data make it impossible to posit a very specific model, the authors should at least give a cogent set of possibilities. As is, there is an implication in the paper that Drep2 may function like mGluR acutely and this is misleading*.

We have worked heavily on the discussion and hope that our hypothesis for the function of Drep-2 has now become clearer. We did not intend to suggest that Drep-2 interacts physically with mGluR (but, apparently, did unwillingly so) and did not stress enough that we could not rescue mutant deficits acutely in adults. We are confident that we have now made these points clear in the revised version of the manuscript.

Reviewer #1:

*In this manuscript the authors present a very extensive characterization of mutants in Drep-2, a protein initially thought to be involved in apoptosis, which now they convincingly show to be associated with synapses and at least in the adult mushroom bodies resides on the postsynaptic side. The authors convincingly show that the protein has a role in glutamatergic signaling-dependent olfactory learning and memory*.

*Nevertheless there are multiple issues to be resolved to improve presentation and above all readability of the manuscript*!

We thank the reviewer for their positive opinion and fully agree that we had to work on the readability of the manuscript.

*1) The manuscript is way too long, unnecessarily in my opinion*.

We have now shortened all parts of the manuscript significantly, particularly the Introduction and Discussion.

*2) The writing style and language need to be improved significantly*.

As outlined above, we went through a sincere effort of improving the style, flow and, importantly, the language of the manuscript.

*3) There is a plethora of data in the supplement that do not add much to the conclusions of the already very large data set*.

We have now condensed the supplemental data substantially. The previous supplemental Figure 1—figure supplement 2 has been included as a main figure (Figure 3); Figure 3–figure supplement 1-3 have been removed.

*4) The Discussion is unnecessarily too long in my opinion*.

We fully agree. We have shortened the Discussion to approximately half of its previous length.

More specifically:

*1) The language and style must be improved because as it is the manuscript does not read well, is full of errors in syntax, colloquialisms and sections appear disconnected as if written in different times by different people. There appears to be no rhyme or reason in the way the Materials and Methods section was assembled. Genetics and husbandry, transgenics, longevity and behavioral methods customarily are placed in proximity, whereas the molecular techniques are grouped together*.

We do apologize for the poor language and writing style. During the work on the manuscript, we changed parts of it so many times that it eventually lost the character of a coherent text. We hope that this issue has been resolved in the revised version. We have also rearranged the Methods, as suggested by the reviewer.

*2)*
Figures 1 and 2
*do not follow the text presentation and in my opinion would benefit the manuscript and help the reader if*
Figure 1
*was made in sequence of panels:*
Figure 1*,*
Figures 1 and 2
Figure 2
*should then be the fractionation (current*
Figure 1*) and*
Figure 1—figure supplement 2*. The text in the results sections should be changed to reflect this*.

We are grateful for this suggestion by the reviewer and have changed figures and text accordingly.

*3) What are the bands around 40kD in the fractionation blot? What are the different fractions? What do the names P1 slof and solid mean? Please describe*.

The bands constitute an unspecific signal (they are still present in the null mutant, see the western blot in Figure 1) that is visible especially clearly under the conditions of this blot. We decided not to cut off the blot above these bands but instead included them as a control, independently validating that we had loaded comparable amounts of protein in each lane. A detailed protocol outlining the procedure of fractionation has now been accepted for publication and is currently in print in *Nature Protocols*: Depner, H., Lützkendorf, J., Babikir, H.A., Sigrist, S.J., and Holt, M.G. (2014): Differential centrifugation-based biochemical fractionation of the Drosophila adult CNS. Nature Protocols, 10.1038/nprot.2014.192.

We now reference this protocol in the main text, the figure legends and the Methods section. We feel that explaining the method in more detail than outlined in the Methods section is beyond the scope of this manuscript. We thus refer now to the aforementioned publication.

*4) Why is counting the number of Kenyon cells in control and mutant animals a measure of apoptosis*???

Given the strong expression of Drep-2 in mushroom body Kenyon cells (KCs) already during larval stages, a potential role of Drep-2 in the regulation of apoptosis might have resulted in altered numbers of KC cell bodies. We agree with the reviewer that this is not a direct measure of apoptosis. However, we do think that an altered count of KCs could be expected if Drep-2 was mainly a regulator of apoptosis. Therefore this measurement is, in combination with the additional data provided in Figure 3, an indication that it is unlikely that apoptosis is misregulated in the CNS of *drep-2* mutants. We have now used more caution in our interpretation of this experiment): “If apoptosis was, nevertheless, misregulated in the CNS of drep-2 mutants in vivo, an altered count of cell bodies should be expected in adult flies. Drep-2 staining was especially prominent at KC synapses in the mushroom body (MB) calyx of wildtypes (Figure 2). We, therefore, quantified the numbers of cell bodies of a subset of MB-intrinsic neurons (KCs). No differences between drep-2 mutants and controls could be observed (Figure 3).”

*5) Why use mushroom body driver c305a with its limited coverage of MB neurons rather than any other one including mb247 which the authors use for many other applications*?

We did the experiment to demonstrate that KC-expressed Drep-2 accumulates at PSDs of KCs within the MB calyx, in the same manner as the endogenous protein does. C305a-Gal4 drives expression in the subset of α’β’-KCs. We have demonstrated in a previous analysis, that these KCs are exclusively postsynaptic in the MB calyx and do not form presynaptic specializations here (Christiansen, F., Zube, C., Andlauer, T.F.M., Wichmann, C., Fouquet, W., Owald, D., Mertel, S., Leiss, F., Tavosanis, G., Farca Luna, A.J., Fiala, A., and Sigrist, S.J. (2011): Presynapses in Kenyon cell dendrites in the mushroom body calyx of *Drosophila*. Journal of Neuroscience 31, 9696–9707.). We have also shown that mb247-Gal4, by contrast, drives expression within different subtypes of KCs, which, in addition to postsynapses, also form presynaptic specializations (active zones) in the calyx. We, therefore, preferred using c305a-Gal4 here to simply avoid any possible confusion between pre- and postsynaptic specializations.

*6) Although it is very nice data, how does inclusion of all the ethanol sensitivity data add to a paper dealing with learning and memory, at least according to its present title*?

We agree with the reviewer and have removed these data, as they are not needed for the main conclusions and the flow of the manuscript.

*7) Shouldn't the calcium imaging data be part of the main figures of the manuscript. They present a significant point, but if part of the main manuscript the effect of the mutation must be rescued*.

We also agree with the reviewer here. Unfortunately, generation of rescue constructs for this experiment (in an isogenized background) would require a significant amount of time. In order not to delay resubmission of the manuscript by a long time, we have decided to remove this dataset from the manuscript. We will consider conducting optimally suited functional imaging experiments on *drep-2* mutants in the future. We hope that the reviewer will understand this decision.

Reviewer #2:

*The ethanol sedation phenotype was rescued, but I had some concerns. In the description of this experiment, the authors focused on Drep-2 expression in the ring neurons of the ellipsoid body—the site of the homer requirement in ethanol sedation and tolerance. However, they never show that the Drep-2 is required in these neurons. When discussing the absence of an interaction with mGluRA, the authors should also cite the possibility that Drep-2 may be required outside of the ring neurons for normal levels of sedation. It is common to perform ethanol sedation experiments separately in males and female as the differences in body size lead to large differences in the rate of sedation. As far as I can tell, this was not done and there is no indication of whether the number of females/males was controlled between genotypes. The rate of sedation was so quick, that there may be a floor effect for the drep-2 mutants, which could confound the ability of ACPD to rescue this phenotype*.

We have also removed ethanol sedation measurements from the manuscript, as suggested by several reviewers and the editor. Of note, only female flies were analyzed in the ethanol sensitivity experiments.

*Also, I am unclear of the status of the white gene in the mutant genotypes. The level of white activity can dramatically impact the rate of sedation (Chan, et al. 2014. Alcoholism: Clinical and Experimental Research 38:1582-1593). It is highly unlikely that the rescue is due to increasing the copies of mini-white, but the authors need to indicate whether the mutants are in a w1118 or w+ background. Also the authors need to justify using frozen fly weight and not protein content as was done in Moore, et al., to normalize ethanol content in the absorption studies. Although, these experiments on ethanol sedation are potentially interesting and could add value to this manuscript, they are not necessary*.

Although these data have now been removed, we still want to point out that we did account for the genetic background of our mutants. We agree that the genetic situation is somewhat confusing. Flies mutant for *drep-2* are, in general, in a *w*^*1118*^ background. However, they carry a *mini-white* insertion on the second chromosome as a remnant of the p-elements used for generating the mutant. Therefore, *drep-2* mutants have red eyes. To account for this situation, we have generated a wildtype control strain that carries a *white*^*+*^ gene but is otherwise in a *w*^*1118*^ background. We included two sets of controls for ethanol sedation experiments: one with white eyes and one with red eyes. Red-eyed controls showed lower ethanol sensitivity than white-eyed controls, but red-eyed mutants showed a higher sensitivity than both controls. Therefore, we were confident that the increased sensitivity of the mutants was not due to the *white* gene.

Reviewer #3:

*This is a dense manuscript, and a schematic outlining the expression patterns of the various proteins (drep-2, mGluR, dfmr), as well as showing the hypothesized interactions between them would significantly help the reader to digest the conclusions. While the nature interaction between drep-2 and mGluR signaling is not completely worked out, showing the directionality of the effects of each manipulation on learning would help*.

We have seriously considered including a figure outlining our model. However, while we believe that our hypothesis of a functional interaction between Drep-2, DmGluRA and FMRP is sound, the details of this interaction are still rather vague. Therefore, as much as we would like to, we do not think that we should provide a model right now, which would, to the casual reader, suggest a level of certainty on the molecular interactions that we do not have yet. However, if the reviewers continue considering inclusion of a model or scheme as helpful to the reader, we would be happy to provide the following model (Figure 8), which summarizes our current hypothesis:Author response image 1.

*Figure S1 shows some nice imaging data suggesting that the MB responses to odorants are attenuated. In panel C, error bars are missing for the drep-2 trace during the odor presentation window (in my copy, at least). Also, the caption should note the location of the recordings (tip of the alpha lobe)*.

We thank the reviewer for spotting this formatting mistake.

*In the last paragraph of the Introduction, the authors abruptly break from discussing previous findings into a discussion of the conclusions in the current manuscript. The last 3-4 sentences are better suited to the Results and Discussion. The authors could replace them with something to the effect of “Here we have examined a potential synaptic role for Drep-2, a CIDE-N domain protein, and its role in regulating learning and memory through interaction with mGluR signaling”*.

We agree with the reviewer and have changed the Introduction accordingly.

*The authors state: “Thus, we obviously produced two null situations for drep-2 and proved the specificity...”. This sounds odd (the “proved” part); I suggest changing to something along the lines of “Thus, the drep-2C-Term antibody demonstrated specificity in both western blots and immune staining. Based on the lack of detectable expression, we conclude that both mutants are null alleles*.”

We have adapted the text accordingly.

*Also: “The adenylyl cyclase Rutabaga mediates coincidence detection between the conditioned and the unconditioned stimulus in KC γ-lobes (Qin et al., 2012)...” Rut-dependent coincidence detection has been directly observed in the alpha lobe by two groups (Tomchik and Davis, 2009; Neuron 64: 510-521; Gervasi et al., 2010; 65: 516-529); these papers should be cited here, particularly since the alpha lobe was imaged*.

During the process of condensing our manuscript and the removal of imaging data, we have removed this sentence.

Reviewer #4:

*The main problem I have with this manuscript is the lack of actual molecular insight into Drep2 function. The data on dfmr and mGluRA are suggestive, but the story is clearly not straightforward and trying to make it fit the dfmr/mGluR template just does not work; it only serves to highlight the fact that something else is going on and the lack of concrete mechanism. There are several pieces of data that lead to sinificant questions that are not answered in this paper*.

*A) mGluRA is not present in the mass spec data set. This does not support a direct adult interaction*.

*B) The data showing normal mGluR levels in Drep2 mutants is also not consistent with direct interaction*.

*C) The fact that mGluR agonists can only influence the Drep2 phenotype if present during development is a huge red flag. If there was a direct interaction, one should be able to get acute interaction. This smacks of some either circuit-level or pathway-level compensation that could be very indirect*.

We agree with the reviewer that we did not find any evidence for a direct interaction of Drep-2 and DmGluRA. On the contrary, based on our data, such a physical interaction of both proteins is rather unlikely. We apologize that, apparently, the presentation of data and formulations within the text nevertheless gave the impression that we suggested such an interaction. We hope that we could clearly make the point in the revised version of the manuscript that we do not believe that Drep-2 interacts directly with DmGluRA. However, we think that our data suggests that Drep-2 stimulates a component of a pathway downstream of DmGluRA.

*D) There is no evidence that the interaction between mGluR and Drep2 occurs in KCs. The authors use drugs and this makes things very hard to interpret. They cite the Kanellopoulos paper which uses ACUTE mGluR RNAi to demonstrate interaction in KCs with dfmr. This cannot be extrapolated to Drep2 because of the developmental nature of that interaction. Again, the issue of circuit level compensation becomes important*.

We have now tried to make it clear throughout the manuscript now that we could only rescue the deficits of *drep-2* mutants when we stimulated mGluRs starting already in development and that this is not the same situation as described by Kanellopoulos *et al.*

*E) The courtship data do not necessarily suggest a “learning” interaction of dfmr and Drep2. Dfmr males do not court and if males do not court in this paradigm they cannot learn since there is an operant component to this behavior. Drep2 mutants do court and loss of Drep2 in the double rescues courtship. The fact that the dfmr;Drep2 males now court could be the basis of the rescue, not an enhancement of learning*.

The reviewer is absolutely right. We nevertheless thought that reporting the functional antagonism between FMRP and Drep-2 in courtship conditioning is interesting. In response to the concerns of the reviewers we have decided to remove the courtship data entirely.